# Spatial assessment of advanced-stage diagnosis and lung cancer mortality in Brazil

**Kálya Yasmine Nunes de Lima**[1], **Marianna de Camargo Cancela**[2], **Dyego Leandro Bezerra de Souza**[3,4] *

**1** Postgraduate Program in Collective Health, Federal University of Rio Grande do Norte–UFRN, Natal, Rio Grande do Norte, Brazil, **2** Division of Population Research, Division of Population Research, National Cancer Institute (INCA), Rio de Janeiro, Brazil, **3** Department of Collective Health, Postgraduate Programme in Collective Health, Federal University of Rio Grande do Norte–UFRN, Natal, Rio Grande do Norte, Brazil, **4** Faculty of Health Science and Welfare, Research group on Methodology, Methods, Models and Outcomes of Health and Social Sciences(M3O), Centre for Health and Social Care Research (CESS), University of Vic-Central University of Catalonia (UVic-UCC), Vic, Spain

* dysouz@yahoo.com.br

**Data Availability Statement:** The data used in the research were extracted from public and open access information systems, which were cited in the manuscript and can be accessed through the

## Abstract

The high incidence and mortality rates make lung cancer a global public health issue. Socio-economic conditions and the provision of health services may be associated with this reality. This study investigates the spatial distribution of advanced-stage diagnosis and mortality due to lung cancer and its association with the healthcare services supply and demographic and socioeconomic indicators in Brazil. This is an ecological study with 161 Intermediate Regions of Urban Articulation. Mortality data were extracted from the Mortality Information System, and the cases of lung cancer were obtained from the Integrator of Hospital-Based Cancer Registries from 2011 to 2015. Analyses employed Moran's I, local indicators of spatial association, and the multivariable model. The proportion of advanced-stage diagnosis was 85.28% (95% CI 83.31–87.10) and was positively associated with the aging rate (Moran's I 0.11; p = 0.02), per capita income (Moran's I 0.05; p = 0.01) and negatively associated with Gini Index (Moran's I -0.16; p = 0.01). The mean age-adjusted mortality rates was 12.82 deaths/100,000 inhabitants (SD 5.12). The age-adjusted mortality rates for lung cancer presented a positive and statistically significant spatial association with all demographic, socioeconomic and healthcare services supply indicators, except for the "density of family health teams" (Moran's I -0.02 p = 0.28). The multivariable model for the mortality rates was constituted by the variables "Density of facilities licensed in oncology", "Per capita income", and "Health plan coverage". The per capita income presented positive association and health plan coverage negative association with age-adjusted mortality rates. Both associations were statistically significant. The variable density of facilities licensed in oncology showed no significant association with age-adjusted mortality rates. There is a high proportion of advanced-stage diagnosis across the Brazilian territory and inequalities in lung cancer mortality, which are correlated with the most developed areas of the country.

links below: https://datasus.saude.gov.br/ https://irhc.inca.gov.br/RHCNet/.

**Funding:** This study was financed in part by the Coordenação de Aperfeiçoamento de Pessoal de Nível Superior – Brasil (CAPES) – Finance Code 001". The funders had no role in study design, data collection and analysis, decision to publish, or preparation of the manuscript.

**Competing interests:** The authors have declared that no competing interests exist.

## Introduction

Global statistics indicate that lung cancer is the second most common type of cancer in the world, following breast cancer, with 2.2 million new cases and 1.8 million deaths estimated for 2020. This corresponds to 11.4% of all diagnosed cancers and 18.0% of all deaths from this cause [1].

In Brazil, in 2019, there were 16,661 deaths from lung cancer in men and 12,593 deaths in women, which correspond to an age-adjusted mortality rate of 16.19 per 100,000 men and 9.84 per 100,000 for women [2]. The most recent assessment of the Brazilian National Cancer Institute estimated that, for the period 2020–2022, lung cancer is the fourth most incident type of cancer in women and the third in men [3].

In most countries, the survival of patients with lung cancer at 5 years after diagnosis is only 10% to 20% [4], due in part to advanced-stage diagnosis. These data characterize lung cancer as one of the most lethal among malignant tumors [1].

Lung cancer staging at diagnosis and its mortality are associated with the histological type, socioeconomic conditions of the population, and the availability and quality of health services. The importance of each factor, however, is highly dependent on regional contexts [5, 6].

In countries with a high Human Development Index (HDI), there are high incidence and mortality rates for lung cancer [1] and an elevated proportion of advanced-stage diagnosis—especially in low-income populations that live in regions with a lower supply of health services for primary prevention, diagnosis, and timely treatment [5, 7]. In countries with low HDI, lung cancer presents lower incidence and mortality rates (compared with high HDI countries) [1], however with a higher proportion of advanced-stage diagnosis, even in higher-income regions and availability of health services [8].

When considering socioeconomic differences, as well as the distribution of health services in Brazilian regions [9], it is necessary to comprehend how these factors determine the stage of diagnosis and mortality from lung cancer. This study investigates the spatial distribution of advanced-stage diagnosis and mortality due to lung cancer and its association with the health-care services supply and demographic and socioeconomic indicators in Brazil.

## Materials and methods

### Study design and spatial analysis units

This is an ecological study that analyzes 161 Intermediate Regions of Urban Articulation (IRUA), defined by the Brazilian Institute of Geography and Statistics (IBGE) in 2013.

The regional division of Brazil consists of twenty-seven Federation Units (FU) grouped into five geographic regions: North, Northeast, Southeast, South, and Midwest. The diversity of each region and the geographic, social, economic, and political changes that occurred in the last three decades led to the necessity of updating the territorial division, which originated the Urban-Regional Divisions. This delimitation considers the organization of the urban network, hierarchical classification of urban centers, and the flows of people and management [10] (S1 Fig).

IRUA concentrate a set of municipalities in the provision of highly complex goods and services, including health services [10]. In this regional division, the territory is organized into metropolises, regional capitals and smaller urban centers, according to municipal boundaries. The distribution of public and private services (such as healthcare, education, security, among others) is considered along with the mobility of the population in search of these services and the regions of influence of the cities, without necessarily following the limits of the FU [11].

The IRUA constitute a territorial mesh defined based on common characteristics and relationships between municipalities, based on an urban center. This division reflects the context of regions and urban articulations established in the period studied [10], which cannot be portrayed using the delimitation of each individual municipality only. For this reason, the IRUA were chosen as the unit of analysis in this study. Access to this digital territorial mesh is public and freely available at the IBGE website [12].

## Study variables and data sources

The response variables were the proportion of advanced-stage diagnosis and age-adjusted mortality rates for tracheal, lung, and bronchial cancer, combined for men and women, by IRUA, for 2011–2015. The expression *tracheal, lung, and bronchial cancer* was reduced to lung cancer to maintain the fluidity and objectivity of the text.

All data included here were collected by municipality and then aggregated to an IRUA territory level by means of averaging. The cases of lung cancer were collected for men and women, aged between 18 and 99 years diagnosed in the period from 2011 to 2015, from the Integrator of Hospital-Based Cancer Registries (Integrator-HBCRs) [13]. For the study period, the Integrator consolidates information from 273 hospital information units installed in general or specialized cancer hospitals of public, private or philanthropic origin [14]. The coverage of HBCRs data is over 70% in the South and North, 68% in the Southeast, 62% in the Northeast and 50% in the Midwest [15].

Based on the Tumor, Nodes, Metastasis (TNM) Classification of malignant tumors, lung cancer cases were classified as advanced-stage (TNM III and IV) and early stage (TNM I and II) [16]. The proportions of advanced-stage diagnosis of lung cancer were then calculated per IRUA. Cases of In Situ disease (TNM 0), with stage not known and without data of residence address were excluded. Fig 1 depicts the cases selection process.

The deaths that occurred in Brazil between 2011 and 2015 due to lung cancer (ICD-10, C33-34) [17] were collected from the Mortality Information System (MIS) [18], for residents of each Brazilian municipality, per age group and sex, for each year. Deaths with no specification of residence and age group were excluded from the study.

For the correction of the number of deaths, the methodology proposed by Santos and Souza [19] was followed, considering redistribution per sex, age group, completeness of

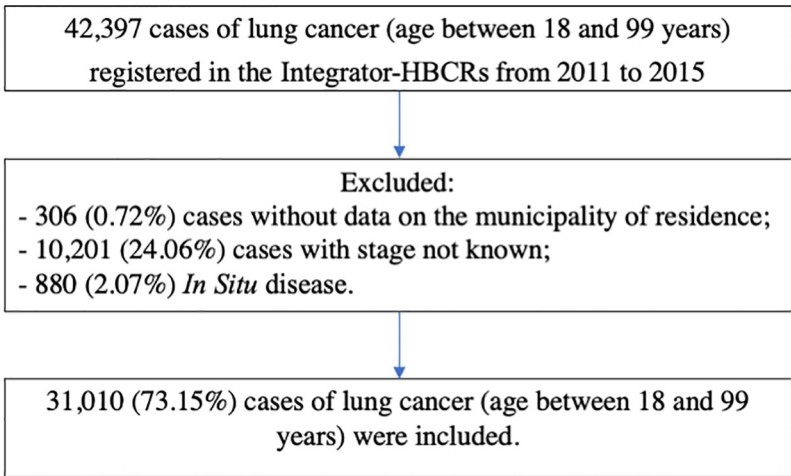

**Fig 1. Flowchart of the selection procedure of lung cancer cases between 2011 and 2015 in the Integrator-HBCRs.**

death record, and ill-defined deaths. Firstly a correction factor was calculated for the analyzed period, considering age group, sex, and FU, from the percentage difference between the number of deaths reported to MIS and the number of deaths redistributed based on chapter II (neoplasms) of the ICD-10. The value 1 was added to the difference obtained to obtain the correction factor per FU, as MIS only provides redistributed data for this territorial level. Finally, the correction factor is multiplied by the number of deaths in each municipality, across all Brazilian FU [19].

The explanatory variables were classified into demographic and socioeconomic indicators (per capita income, aging rate, Gini Index and urbanization degree) and indicators related to the healthcare services supply (health plan coverage, density of general practitioners, density of facilities licensed in oncology, density of diagnostic equipment, density of specialist doctors, and density of family health teams).

Demographic and socioeconomic variables for 2010 were obtained from the Brazilian Atlas of Human Development of United Nations Development Programme (UNDP) [20]. Data on the density of medical doctors and healthcare services supply were extracted from the Brazilian Registry of Health Facilities (CNES) [21] and the National Regulatory Agency for Private Health Insurance and Plans (ANS) [22] for 2013. Table 1 presents the study variables and corresponding descriptions.

The crude and age-adjusted mortality rates for lung cancer (AMR) (per 100,000 inhabitants) were calculated for each IRUA. Age-adjusted rates were calculated from the direct standardization method [23], using the world standard population as reference [24]. The 2013 population was used as a reference for the calculation of indicators along with data on municipality, sex and age, as published by IBGE [25]. Table 1 presents the study variables and corresponding descriptions.

## Statistical analysis

Software TerraView 5.0.0 [26] was used for the descriptive analysis and creation of the distribution maps of the proportion of advanced-stage diagnosis and age-adjusted mortality rates for lung cancer. The criterion used to define the categories of quantitative maps was equal intervals. The Global Moran Index was used to verify the spatial autocorrelation of proportions of advanced-stage diagnosis and age-adjusted mortality rates for lung cancer. The presence of spatial clusters was analyzed using the Local Indicator of Spatial Association (LISA).

According to the LISA significance level, in the spatial autocorrelation analysis the regions were classified as high-high when the area is formed by IRUA with a high frequency of the variable and also surrounded by high-frequency IRUAs; low-low when the area is formed by IRUA with low frequency variable and also surrounded by low frequency IRUAs; high-low when IRUA with high frequency of the variable is surrounded by regions of low frequency, and low-high when IRUA with low frequency of the variable is surrounded by high frequency IRUAs.

LISA bivariate analysis was performed within the GeoDa 1.6.61 software [27] to assess the spatial association between the outcomes studied (proportion of cases diagnosed at an advanced stage and age-adjusted mortality rates for lung cancer) and each explanatory variables. The analysis generated the Moran Local index, maps and scatter plot of correlations. The statistical significance of Moran's I was verified by a random permutation test, with 99 permutations [27]. Analysis of the surroundings used a first-order Queen Contiguity criterion.

There are five types of spatial clusters in the bivariate spatial association: not significant (territories that did not enter the formation of clusters, because their differences were not significant); high-high (areas formed by IRUAs with high frequencies of the explanatory variable and

**Table 1. Characteristics and details of response variables and explanatory variables for spatial analysis of mortality and advanced-stage diagnosis of lung cancer in Brazil, 2011–2015.**

| Variable | | Source | Description |
|---|---|---|---|
| **Response variable** | **Adjusted mortality rates for lung cancer** | MIS Data from 2011 to 2015 | Lung cancer mortality rate, combined for men and women, adjusted for age based on standard world population. |
| **Response variable** | **Advanced-stage diagnosis of lung cancer** | Integrator-HBCRs Data from 2011 to 2015 | Proportion of advanced-stage diagnosis of lung cancer considering the TNM System for Tumor Classification (TNM III and IV) |
| **Explanatory Variables (Contextual)** | **Per capita income** | Brazilian Atlas of Human Development (2010) | Ratio between the sum of the income of all residents of permanent private households and the total number of residents, per IRUA. |
| | **Gini Index** | | Measures the degree of inequality in the distribution of individuals according to the per capita household income |
| Socioeconomic and demographic | **Aging rate** | | Ratio between the population aged 65 and over and the total population, per IRUA |
| | **Urbanization degree** | | Ratio between the number of people residing in urban areas in relation to the total population of the space where this urban area is located, multiplied by 100, per IRUA. |
| **Explanatory Variables (Contextual)** | **Density of General Practitioners** | CNES (January-December 2013) | Ratio between the average number of general practitioners registered by CNES in 2013 and the total population, multiplied by 100,000, per IRUA. |
| Density of professionals and healthcare services supply | **Density of medical specialists in oncology** | | Ratio between the average number of medical specialists in oncology registered by CNES in 2013 and the total population, multiplied by 100,000, according to IRUA. |
| | **Density of diagnostic equipment -X-ray and computed tomography** | | Ratio between the average number of radiography and computed tomography devices in use registered by CNES in 2013 and the general population, multiplied by 1,000,000 inhabitants, according to the IRUA. |
| | **Density of facilities licensed in oncology** | | Ratio between the average number of facilities licensed in oncology registered by CNES in 2013 and the general population, multiplied by 1,000,000 inhabitants, according to the IRUA |
| | **Density of family health teams** | | Ratio between the average number of family health teams registered by CNES in 2013 and the general population, multiplied by 100,000, per IRUA. |
| | **Health plan coverage** | ANS (January-December 2013) | Average of the ratio, expressed in percentage, between private health insurance coverage and the total population, per IRUA. |

MIS: Mortality Information System;

Integrator-HBCRs: Integrator of Hospital-Based Cancer Registries

TNM: Tumor, Nodes, Metastasis.

CNES: Brazilian Registry of Health Facilities

ANS: National Regulatory Agency for Private Health Insurance and Plans

IRUA: Intermediate Regions of Urban Articulation

high frequencies of the response variable); low-low (areas formed by IRUAs with low frequencies of the explanatory variable and low frequencies of the response variable); high-low (areas formed by IRUAs with high frequencies of the explanatory variable and low frequencies of the response variable); and low-high (areas formed by IRUAs with low frequencies of the explanatory variable and high frequencies of the response variable).

The multivariable analysis of lung cancer mortality included explanatory variables that showed a statistically significant association with the response variable and that were non-collinear with each other (correlation < 0.7). The spatial error, classic, and spatial lag regression models were compared (S1 Table), and the global spatial effect model selected for the study was the spatial error model. The final model was selected based on the highest value of the log-likelihood, lowest values of the Akaike Information Criterion (AIC) and Schwarz information criterion [28], and on theoretical plausibility.

The spatial autocorrelation of residues of the multivariate model was assessed after defining the multivariate model, using Moran's I and data scatter plot.

This study used secondary data collected from health information systems, which are open and freely accessible. These systems do not provide individual identification data and therefore the approval by a Research Ethics Committee (CEP) was not required, following Resolution 580/2018 [29].

## Results

Between 2011 and 2015, 31,010 lung cancer cases were diagnosed (with staging data and place of residence), 12,525 in women and 18,485 in men. The Proportion of Advanced-Stage Diagnosis for Brazil was 85.28% (95% CI 83.31–87.10) for the population between 18 and 99 years of age. For women, the proportion of advanced-stage diagnosis was 84.08% (95% CI 81.65–86.51), and for men, 85.52% (95% CI 83.26–87.77).

The IRUAS of Tefé and Tabatinga, located in the North region, had no records of cases of lung cancer diagnosed at an advanced stage (proportion of 0%). The IRUAS of Juazeiro do Norte, Ararapina, Guarabira, Afogados da Ingazeira (located in the Northeast region), Lavras (Southeast region) Parintins (North region), and Cáceres and São Félix do Araguaia (Midwest) presented a proportion of 100% for cases diagnosed in stage III or IV. Proportion of advanced-stage diagnosis for both sexes presented low spatial autocorrelation, but significant (Moran's I 0.37; p = 0.01). The formation of only one low-low cluster located in the North region was observed.

After correction, 29,371 deaths from lung cancer were added to the initial number, which corresponds to 19.36% of the deaths included in the study. The number of deaths resulted in 151,699: 96,758 in men and 54,941 in women. The AMR for lung cancer in Brazil was 8.44 (SD 3.14) per 100,000 women and 19.25 (SD 13.83) per 100,000 men. The combined AMR for men and women was 12.82 (SD 5.12) deaths per 100,000 inhabitants.

The Itaberaba and Bom Jesus da Lapa IRUAS, located in the Northeast region, presented the lowest age-adjusted mortality rates for lung cancer, 3.95 and 4.95 deaths per 100,000 inhabitants, respectively. The highest age-adjusted rates were verified in the IRUAS of Brasília (39.68 deaths per 100,000 inhabitants), in the Midwest region, and Santa Cruz do Sul (29.49 deaths per 100,000 inhabitants), in the South region of Brazil.

AMR presented significative spatial autocorrelation (Moran's I 0.50, p = 0.01) with the formation of Low-low clusters, especially in the North and Northeast regions, and High-high clusters in the South and Midwest, as depicted in Fig 2.

Demographic and socioeconomic explanatory variables demonstrated weak spatial association with advanced -stage diagnosis, with statistical significance for Gini Index (Moran's I -0.16, p = 0.01), aging rate (Moran's I 0.11, p = 0.02), and per capita income (Moran's I 0.05, p = 0.01) (Fig 3).

Regarding the variables related to the healthcare services supply, no statistically significant associations were observed with proportion of advanced-stage diagnosis, as displayed in Fig 4.

Figs 5 and 6 show the spatial association between AMR, for both sex, for lung cancer and the demographic and socioeconomic indicators and healthcare services supply indicators. All associations were statistically significant, except for the variable "density of family health teams" (Moran's I -0.02 p = 0.28). The AMR were not associated with proportion of advanced-stage diagnosis (Moran's I -0,04 p = 0.18) (S2 Fig).

In the multivariable spatial regression analysis, the spatial error model presented the best fit for the studied variables (S1 Table). The explanatory power of the final model was 66.3% for lung cancer mortality. The residues presented normal distribution and Moran's I was -0.04

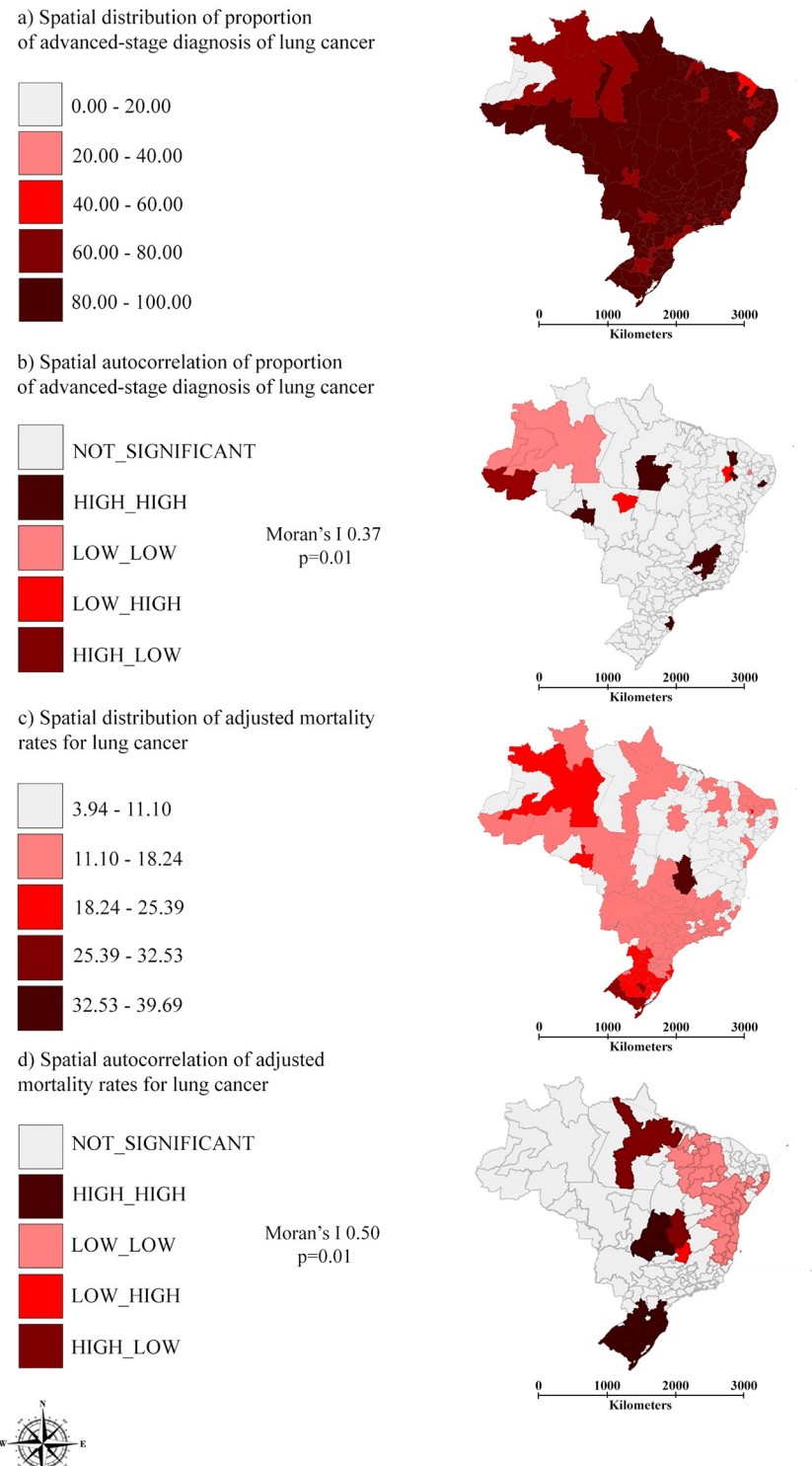

**Fig 2.** (a) Spatial distribution and (b) spatial autocorrelation of proportion of advanced-stage diagnosis, for both sexes, of lung cancer; (c) spatial distribution and (d) spatial autocorrelation of age-adjusted mortality rates for lung cancer, for both sexes per IRUA, Brazil, 2011–2015.

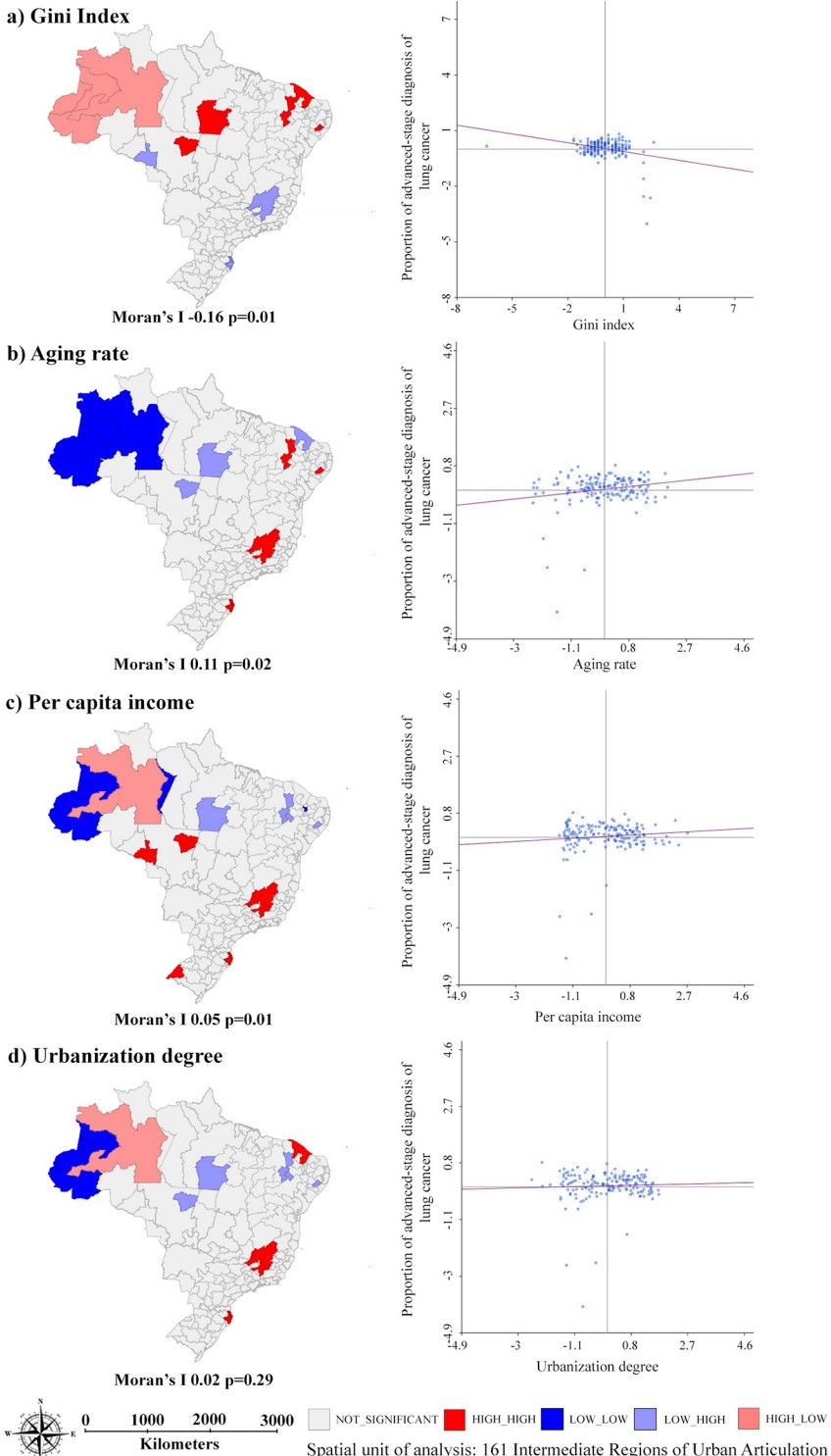

**Fig 3. Spatial association between proportion of advanced-stage diagnosis of lung cancer, for both sexes, and demographic and socioeconomic indicators per IRUA Brazil, 2011–2015.**

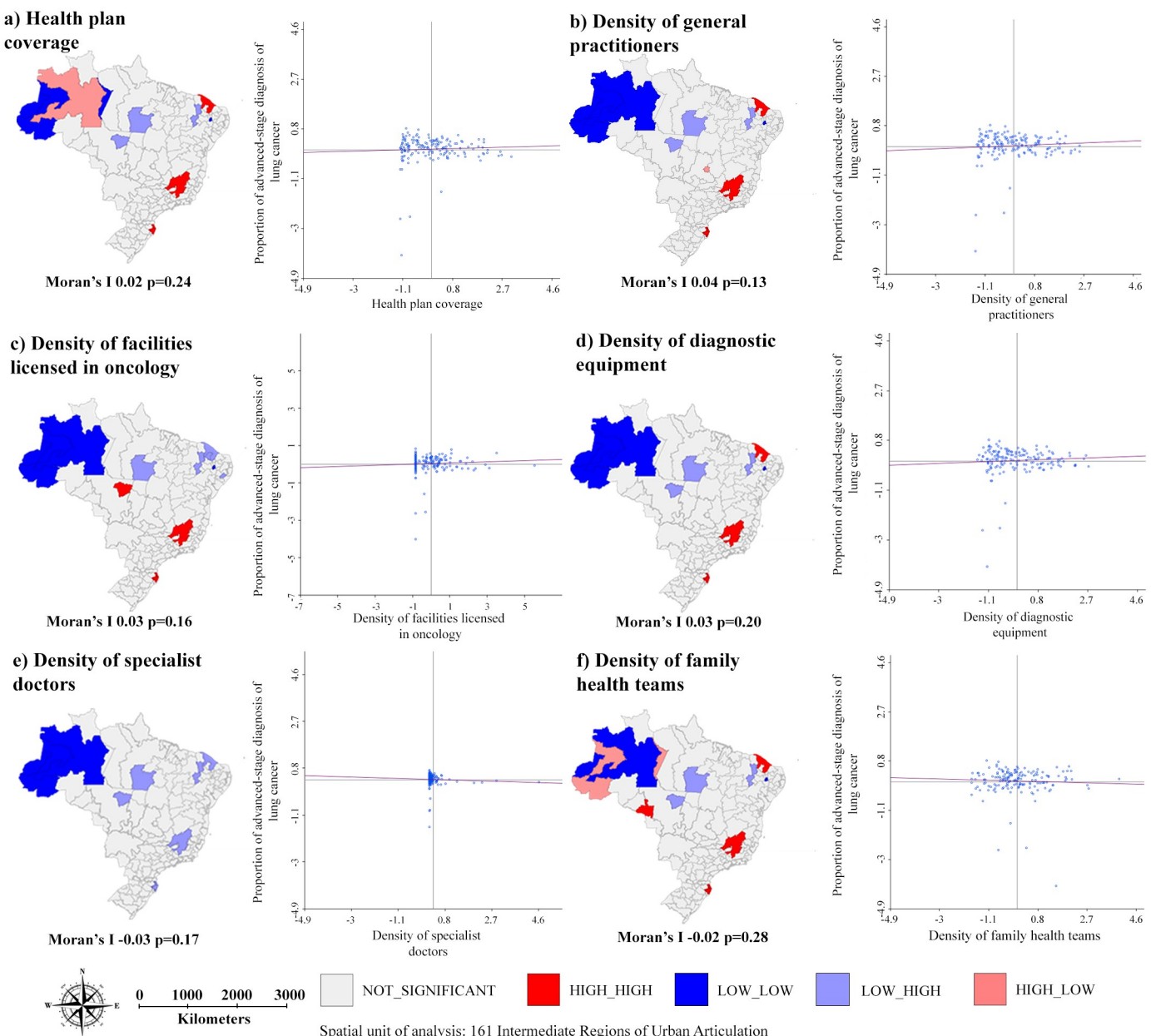

**Fig 4. Spatial association between proportion of advanced-stage diagnosis of lung cancer, for both sexes, and healthcare services supply indicators per IRUA, Brazil, 2011–2015.**

(p = 0.29) (S3 Fig). The variables that remained in the model were "density of facilities licensed in oncology", "Per capita income", and "health plan coverage". Table 2 presents the data of the spatial regression analysis for age-adjusted mortality rates for lung cancer.

The per capita income variable was positively associated and health plan coverage was negatively associated with age-adjusted lung cancer mortality rates. Both associations were statistically significant. The variable Density of Facilities Licensed in Oncology showed no significant associated with age-adjusted lung cancer mortality rates. The choice to keep this variable in the final model considered the theoretical plausibility and its ability to adjust the model.

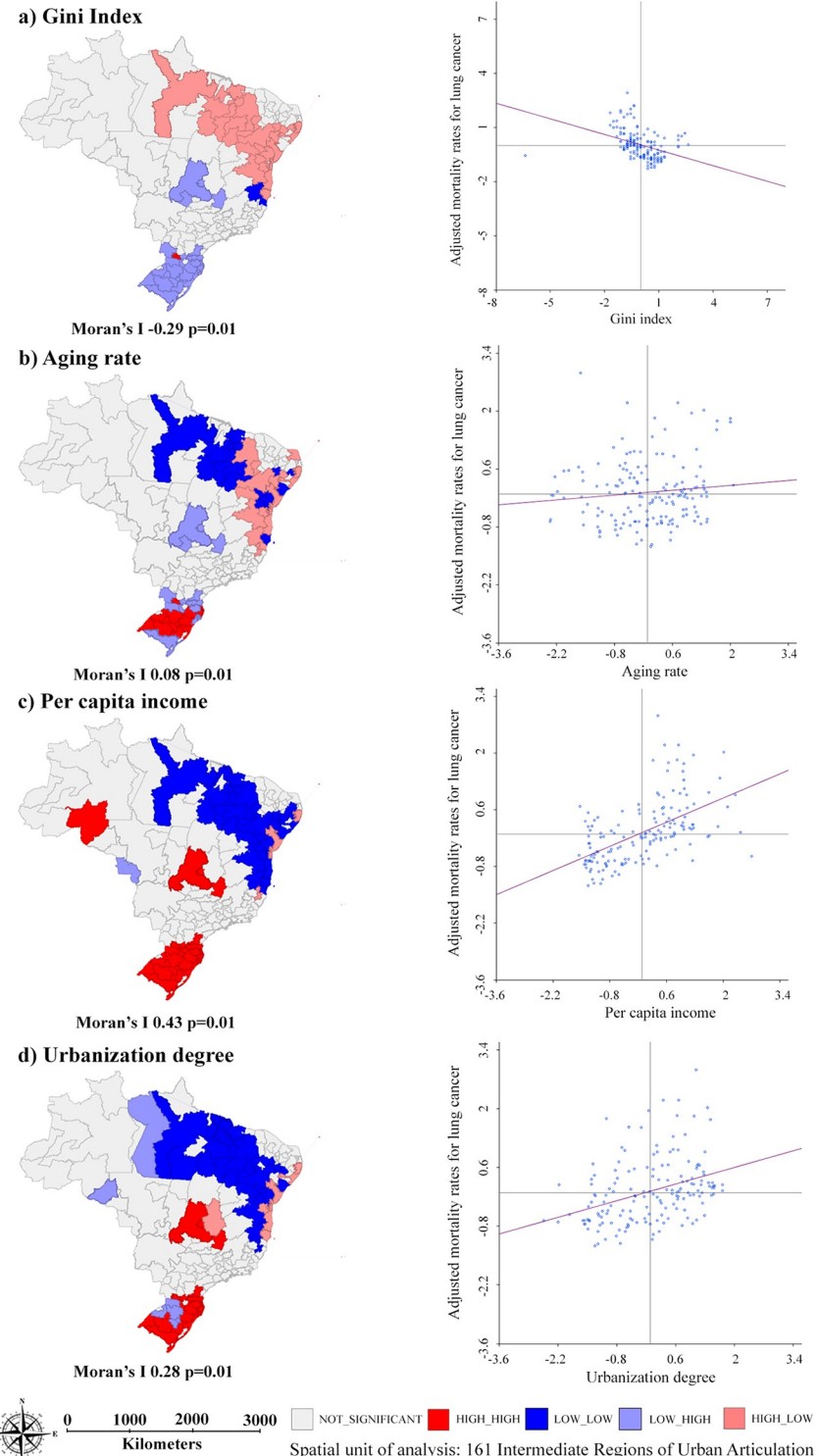

**Fig 5. Spatial association between age-adjusted mortality rates for lung cancer, for both sexes, and demographic and socioeconomic indicators per IRUA, Brazil, 2011–2015.**

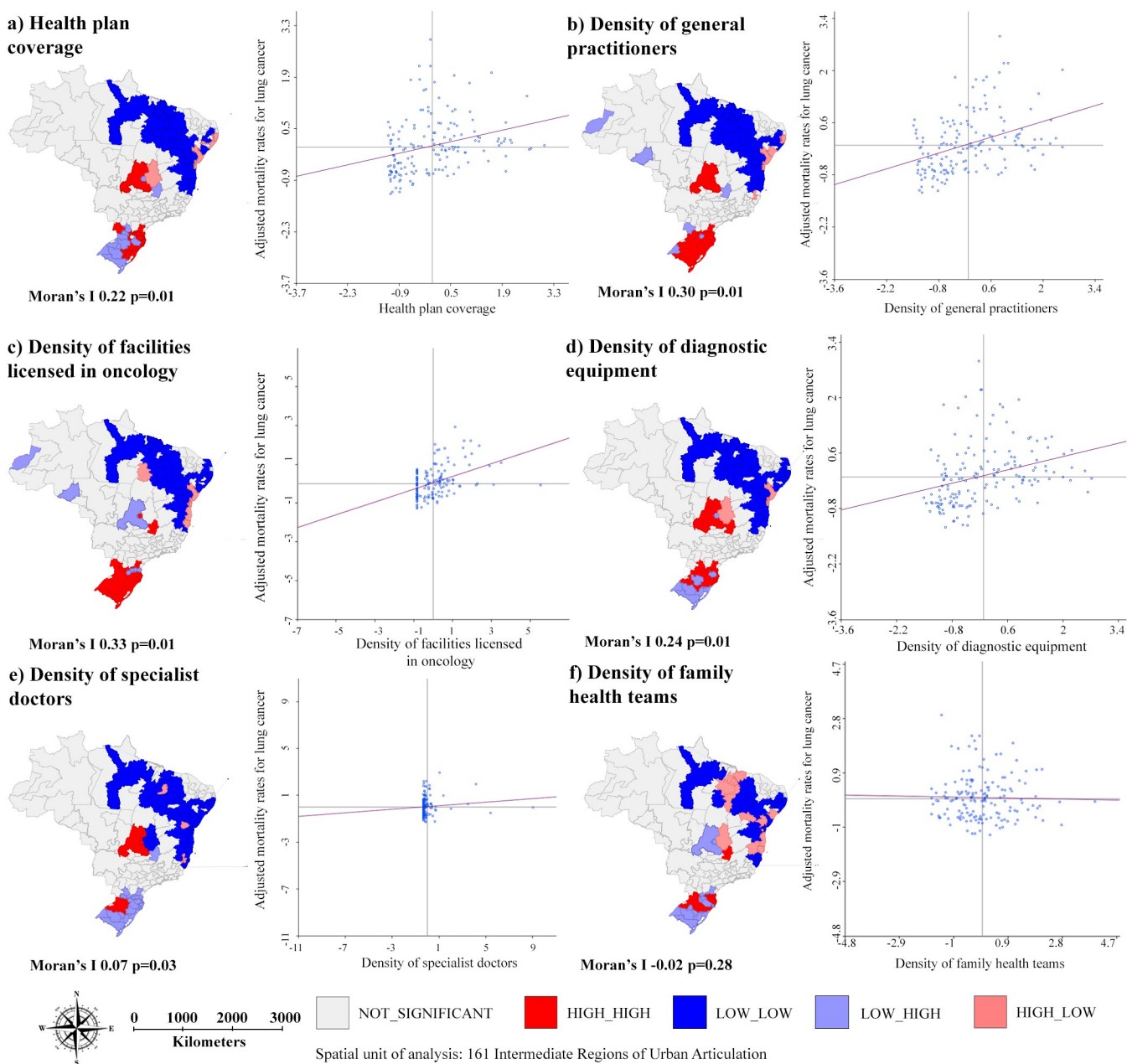

**Fig 6. Spatial association between age-adjusted mortality rates for lung cancer, for both sexes, and healthcare services supply indicators per IRUA, Brazil, 2011–2015.**

## Discussion

The study showed a weak spatial dependence for the proportion of advanced-stage diagnosis in Brazilian IRUAS. There are three IRUAs in the Northeast, two in the North and one in the South and Southeast regions with high proportion of advanced-stage lung cancer diagnoses and one low-low cluster of proportion of advanced-stage lung cancer diagnoses in the North region. Only demographic and socioeconomic contextual variables were correlated with this outcome.

**Table 2. Spatial regression analysis of age-adjusted lung cancer mortality rates and its association with demographic and socioeconomic and healthcare services supply indicators, per IRUA, Brazil, 2011–2015.**

| Variable | Coefficient | Standard error | z-value | p |
|---|---|---|---|---|
| Constant | 2.781 | 1.528 | 1.819 | 0.068 |
| Density of Facilities Licensed in Oncology | 0.518 | 0.130 | 0.396 | 0.691 |
| Per capita income | 0.020 | 0.002 | 7.794 | <0.001* |
| Health plan coverage | -0.163 | 0.051 | -3.195 | <0.001* |
| Lambda | 0.787 | 0.081 | 9.671 | <0.001* |

*Statistically significant;

Model fit ($x^2$): $R^2$ = 0.663

Nevertheless, the age-adjusted mortality rates for lung cancer were more influenced by social contexts and the healthcare services supply, leading to the formation of high mortality clusters in IRUA of the Midwest and South and low mortality clusters in IRUA of the North and Northeast regions.

The proportion of lung cancer cases diagnosed in advanced-stage was high in Brazil (85%), similar to the ratio found in the UK (87%) [30] and India (88%) [8] and higher than for the USA (57%) [31] and Canada (65%) [32].

Although there is literature consensus on the histological types of lung cancer with a higher risk of advanced-stage diagnosis, divergent results are obtained regarding the effect of socio-economic conditions and the provision of health services on the stage of the neoplasia in the diagnosis [5–7].

In the USA, late diagnosis is more prevalent in regions with higher proportions of African-American individuals and lower incomes [5]. In France, patients residing in needier areas are less prone to curative surgery due to the advanced-stage diagnosis of lung cancer [7]. Nevertheless, a systematic review that approached different health system organizations verified a lack of evidence regarding a socioeconomic gradient in lung cancer diagnosis in later stages [6].

In the present study, the advanced-stage diagnosis of lung cancer is prevalent in almost the entire Brazilian territory. The single low mortality cluster identified can be a consequence of the absence of records of diagnosed cases of lung cancer or loss of cases due to lack of cancer staging, during the study period, in the IRUA of Tabatinga and Tefé (Northern region).

A high proportion of advanced-stage diagnoses was verified in IRUAs from different regions, especially those located in more developed areas (South and Southeast), which historically also have higher rates of lung cancer incidence and mortality [33]. The proportion of advanced-stage diagnosis was positively associated with the aging rate and per capita income and inversely associated with the Gini Index.

Introducing territory as an analysis unit requires the understanding that territory is the result of historical socioeconomic and political appropriation. Therefore, the current social and health-related inequalities are the reflexes of decades of social and economic policies that favored the development of some regions to the detriment of others [34].

Due to the continental dimensions of Brazil and its economic growth process, the country presents pronounced social and health-related inequalities. The concentration of industrial growth in the Midwest, South, and Southeast regions during the 20th century led to a division of the territory into two large blocks. The first block, constituted by these developed regions, presenting higher development levels, and the second block, encompassing the North and Northeast regions, with the worst socioeconomic indices [35], as seen in the clusters with low income and low urbanization in the North (Figs 3 and 5) and in the Northeast regions (Fig 5).

This scenario is also affected by the demographic transition that the country has been experiencing throughout the most recent decades, with an increase of older population in more developed regions of Brazil [36].

The association between higher incomes, aging rates, and lower Gini index is verified in territories with higher development indices which, in Brazil, coincide with IRUAs that present better healthcare services supply and older population [33, 36]. The high proportions of lung cancer cases diagnosed at a late stage in more developed regions can reflect a better quality of the registries of cases diagnosed in these territories [37].

The weak associations between the proportions of advanced-stage diagnosis of lung cancer and socioeconomic indicators, and the absence of association with the variables of healthcare services supply, verified herein, corroborate previous research. In India, an association was established between low education levels and advanced-stage diagnosis of breast, uterus, and oral cavity cancers, with no association established for lung cancer [8]. In Forrest et al. study [6], it was verified that in countries with universal health systems (e.g., Denmark and England), such as Brazil, there was no evidence of an association between advanced-stage diagnosis of lung cancer and social inequalities and availability of health services.

For the analyzed period, Brazil presented lung cancer mortality, for both sexes, which were lower than high-income countries, such as Germany (31.9/100,000 men; 14.8/ 100,000 women) and the USA (31.4/100,000 men; 21.3/100,000 women). In comparison with some South-American countries, mortality rates for lung cancer were higher than Colombia (13.1/100,000 men; 6.3/100,000 women) and lower than Argentina (26.5/100,000 men; 9.3/100,000 women) [38].

This study evidenced an unequal distribution pattern of lung cancer mortality in the Brazilian territory. High rates are verified in regions with better socioeconomic and health-related indicators, while the least developed regions concentrate lower rates, also a reality of other national and international studies [1, 36]. High mortality clusters in IRUA of the Midwest and South and low mortality clusters in IRUA of the North and Northeast reinforce this pattern, along with the existence of historical territory-related factors that influence illness and death due to lung cancer, in the period studied.

Lung cancer mortality, in general, is related to the consumption of tobacco, high lethality, and late-stage diagnosis. Regional variations are mainly associated with exposure to the leading risk factor and access to health services for prevention, diagnosis and/or timely treatment of the disease [1, 33].

In Brazil, the production and commercialization of tobacco reached a peak in the 1980s, with higher prevalence in the South region, due to regional characteristics that facilitated tobacco cultivation and the strong influence of European immigrants in the area, regarding cultivation and consumption [33]. Long-term results of this economic activity and its relationships with other sectors of the society include the higher prevalence of tobacco consumption, even nowadays. There are also higher mortality rates for lung cancer in Southern regions, in comparison with the other Brazilian geographic regions [39], as verified herein.

Socioeconomic conditions, especially income and education, are crucial factors determining the illness and mortality of lung cancer [5]. Cancer studies carried out in Brazil [36, 40] in agreement with the results obtained herein, demonstrated a positive association between lung cancer mortality and better socioeconomic conditions. This scenario is also seen in the high lung cancer mortality rates in higher-income countries [41]. High exposure to risk factors and higher rates of aging in regions with better demographic and socioeconomic indicators may justify this fact [33].

Regarding the healthcare services supply, in this study, the high mortality rates were associated with territories that presented low coverage of health plans and high density of facilities licensed in oncology.

In Brazil, health-related services can be accessed through the Unified Health System (SUS, the Brazilian publicly funded universal health system), health plans, or privately. Although there is a universal and free health system, in the last 15 years, there has been a progressive increase in the utilization of health plans, especially in regions with higher income and higher aging rates, located in the Southeast and Midwest [42]. The decision to acquire health insurance usually occurs due to the perceived deficiency of SUS to satisfy the demands of the population. Private health plans guarantee access and assistance only for a specific share of the Brazilian population, usually constituted by wealthier individuals [43].

The inequalities related to the healthcare offered by public and private systems evidence broader assistance to those with health plans. In cancer assistance, the differences are primarily perceivable regarding medical appointments, emergency assistance, and hospitalizations, which are more frequent in patients of the private system [44].

In the South, the research evidenced IRUA with high mortality rates for lung cancer and low coverage of health plans. Although the South region is recognized for its highest socioeconomic indicators, compared with the other Brazilian regions, it still has high levels of social inequalities, mainly in the Rio Grande do Sul FU [44, 45]. Limited access to health services, caused by socioeconomic inequalities, could have contributed to high mortality rates [35, 46].

Moroever, changes in address, in search of treatment, can also move part of the population with higher income to other regions with other health services, including private ones. Likewise, people who need access to health services through the Unified Health System can migrate to places with a greater offer of public services. Both situations can contribute to the reduction of health plan coverage in regions with high mortality rates.

Regarding health units accredited in oncology in Brazil, cancer treatment can be performed in general hospitals licensed by the Ministry of Health, such as the Units of High Complexity Assistance (UNACON) and Centers of High Complexity Cancer Assistance (CACON), tied to SUS. UNACONs can provide specialized healthcare for the most prevalent cancers, which can include or not radiotherapeutic assistance. CACONs offer specialized assistance to all types of cancer and count with standard radiotherapeutic assistance [47].

The availability of these services is still insufficient for the Brazilian reality—in 2017, there were only 44 licensed CACONs in Brazil, of which 70.5% were located in the South and Southeast regions [48], where higher mortality rates were also found. This association between cancer mortality and areas with a more comprehensive offer and availability of health facilities licensed in oncology could result from high number of diagnoses of the disease, more structured surveillance and better quality death records in these places [40, 49]. In addition, migration of people to these regions in search of cancer diagnosis and treatment services is highlighted. These populational displacements can concentrate mortality rates in more developed regions [50].

The high offer of cancer-related services associated with lung cancer mortality can also evidence the quality of the care offered. The study by Kaliks et al [51] identified differences in the quality of treatment for the four most common cancers in Brazil, among the Public Health System (SUS), the assistance provided by health plan and the Therapeutic Guidelines established by the Ministry of Health. Of the 52 SUS cancer treatment centers investigated, only 29 followed the directives regarding lung cancer treatment. Eight presented higher standards, five were compatible, and 16 presented lower standards than suggested by the Therapeutic Directives. Furthermore, 19 SUS cancer centers presented standards below the recommended and practiced by health plans.

These differences emphasize that most treatment centers offer lower-quality care than preconized by the Ministry of Health and provided by health plan. This can contribute to lung cancer mortality, especially in poverty-stricken populations. The existence of differences in the

treatments and therapeutic procedures for lung cancer causes different response patterns and health inequalities. The individual can be treated at different levels (higher or lower) than the standard suggested by the Health Ministry, depending on the location and type of health service access [51].

It must be highlighted that even given the inequalities regarding lung cancer illness and mortality, Brazil is globally recognized for its initiatives of reducing the exposure of the population to the leading risk factor for lung cancer, which is the consumption of tobacco. The implementation of policies to control the cultivation, production, and commercialization of tobacco and the National Program of Tobacco Control [52] have presented positive results. Between 1983 and 2013, tobacco consumption in Brazil reduced from 34.8% to 14.7% [53, 54]. Between 2011 and 2013, it was verified that the share of families abandoning the cultivation of tobacco increased from 6.5% to 10.2% [55].

The analysis shows that the spatial distribution of the proportion of advanced-stage diagnosis of lung cancer in the Brazilian IRUA is weakly related to the demographic and socioeconomic context. For mortality rates, it was possible to verify historical and current territory-related structures that correlated with mortality from this disease. Regional inequalities in mortality rates, due to socioeconomic disparities, demonstrate the complexity in the vigilance of lung cancer in a country that presents a mixture of low- and high-income country characteristics [40].

Limitations of this study include the absence of incidence data for lung cancer per IRUA and of variables that assess other issues related to access to health services. These additional variables, such as the time elapsed until diagnosis and type of entry in the health system, could help to understand the spatial dimension of cases diagnosed at an advanced-stage and lung cancer mortality rates. Some cases of lung cancer may have been missed, being registered only on the death certificate. Cases excluded, due to the absence of data, could present a non-random spatial distribution, which was not assessed herein.

The moderate spatial dependence of the proportion of advanced-stage diagnosis of lung cancer (Moran's I 0.37) and the weak associations with the explanatory variables studied (as seen in Figs 3 and 4) made it impossible to design a multiple analysis model. There is also a limitation related to the use of indicators at aggregated levels, which do not necessarily reflect the reality at individual levels.

## Conclusions

A high proportion of advanced-stage diagnosis of lung cancer was verified for almost the entire Brazilian territory for the study period. The proportion of advanced-stage diagnoses was weakly spatially associated only with socioeconomic and demographic indicators. However, age-adjusted mortality rates for lung cancer presented irregular distribution in the Brazilian IRUA, associated with territories that present higher incomes and lower coverage of health plans, regardless the supply of health facilities licensed in oncology.

The results indicate the necessity of strategies that enable access to primary prevention, diagnosis, and timely treatment to reduce the proportion of advanced-stage diagnosis and inequalities in the mortality of lung cancer across Brazilian regions.

## Supporting information

**S1 Fig. Regional division of Brazil into five geographic regions and 161 Intermediate Regions of Urban Articulation.**
(TIF)

**S2 Fig. Spatial association between age-adjusted mortality rates for lung cancer and the proportion of advanced-stage diagnosis of lung cancer, Brazil, 2011–2015.**
(TIF)

**S3 Fig. Analysis of residues for the spatial error model of lung cancer mortality Brazil, 2011–2015.**
(TIF)

**S1 Table. Models of spatial regression, according to the analysis criteria for selecting the final model.**
(DOCX)

## Author Contributions

**Conceptualization:** Kálya Yasmine Nunes de Lima, Dyego Leandro Bezerra de Souza.

**Data curation:** Kálya Yasmine Nunes de Lima, Marianna de Camargo Cancela, Dyego Leandro Bezerra de Souza.

**Formal analysis:** Kálya Yasmine Nunes de Lima, Marianna de Camargo Cancela, Dyego Leandro Bezerra de Souza.

**Investigation:** Kálya Yasmine Nunes de Lima.

**Methodology:** Kálya Yasmine Nunes de Lima, Marianna de Camargo Cancela, Dyego Leandro Bezerra de Souza.

**Project administration:** Dyego Leandro Bezerra de Souza.

**Resources:** Kálya Yasmine Nunes de Lima.

**Supervision:** Dyego Leandro Bezerra de Souza.

**Writing – original draft:** Kálya Yasmine Nunes de Lima.

**Writing – review & editing:** Kálya Yasmine Nunes de Lima, Marianna de Camargo Cancela, Dyego Leandro Bezerra de Souza.

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
