## [Decision Letter · Decision Letter 0]

3 Sep 2021

PONE-D-21-20112Spatial assessment of advanced stage diagnosis and lung cancer mortality in BrazilPLOS ONE

Dear Dr. de Souza,

Thank you for submitting your manuscript to PLOS ONE. After careful consideration, we feel that it has merit but does not fully meet PLOS ONE’s publication criteria as it currently stands. Therefore, we invite you to submit a revised version of the manuscript that addresses the points raised during the review process.

We look forward to receiving your revised manuscript.

Kind regards,

Edison I. O. Vidal, MD, MPH, PhD

Section Editor

PLOS ONE

Journal Requirements:

2. Thank you for stating the following in the Acknowledgments/ Funding Section of your manuscript: 

This study was financed in part by the Coordenação de Aperfeiçoamento de Pessoal de Nível Superior – Brasil (CAPES) – Finance Code 001”.

This study was financed in part by the Coordenação de Aperfeiçoamento de Pessoal de Nível Superior – Brasil (CAPES) – Finance Code 001”. The funders had no role in study design, data collection and analysis, decision to publish, or preparation of the manuscript.

4. We note that Figures 1, 2, 3, 4, 5, S1 & S2 in your submission contain map images which may be copyrighted. All PLOS content is published under the Creative Commons Attribution License (CC BY 4.0), which means that the manuscript, images, and Supporting Information files will be freely available online, and any third party is permitted to access, download, copy, distribute, and use these materials in any way, even commercially, with proper attribution. For these reasons, we cannot publish previously copyrighted maps or satellite images created using proprietary data, such as Google software (Google Maps, Street View, and Earth). For more information, see our copyright guidelines: http://journals.plos.org/plosone/s/licenses-and-copyright.

a. You may seek permission from the original copyright holder of Figures 1, 2, 3, 4, 5, S1 & S2 to publish the content specifically under the CC BY 4.0 license.  

Additional Editor Comments:

I have read your manuscript with interest and have little to add to the several high-quality comments provided by several reviewers. Please notice that PLOS ONE requires authors to make all data necessary to replicate their findings publicly available. Please read carefully the PLOS policy on data availability (https://journals.plos.org/plosone/s/data-availability) since it is one of the criteria for publication in PLOS ONE.

Reviewers' comments:

Reviewer's Responses to Questions

**Comments to the Author**

1. Is the manuscript technically sound, and do the data support the conclusions?

Reviewer #1: Yes

Reviewer #2: Yes

Reviewer #3: Partly

Reviewer #4: Yes

2. Has the statistical analysis been performed appropriately and rigorously? 

Reviewer #1: Yes

Reviewer #2: Yes

Reviewer #3: No

Reviewer #4: Yes

3. Have the authors made all data underlying the findings in their manuscript fully available?

Reviewer #1: Yes

Reviewer #2: No

Reviewer #3: No

Reviewer #4: No

4. Is the manuscript presented in an intelligible fashion and written in standard English?

Reviewer #1: Yes

Reviewer #2: No

Reviewer #3: No

Reviewer #4: Yes

5. Review Comments to the Author

Reviewer #1: This is a very well-written manuscript based on a carefully methodology related to spatial analysis, and the findings are very clearly presented.

I have some comments about the manuscript that follow below:

Methods:

- Why the educational level was not used as an independent variable in the correlation analysis and multivariate model? It's an appropriate indicator of socioeconomic position. In the fourteen paragraph of the discussion section the authors stated that: "socioeconomic conditions, specially income and education, are crucial factors determining the lines and mortality of lung cancer".

- About the Integrator of Hospital Cancer Records. What is the national coverage? Only lung cancer cases from public health services or covers cases from the private system? Are all units that receive cancer patients included in the integrator?

- How the variables were grouped within the IRUA? By the average? Are the municipal boundaries coincident with the IRUA boundaries?

- The authors used a bivariate correlation analysis. In the statistical analysis I suggest to add the information that Global Moran’s Index and the Local Indicator of Spatial Association (LISA) were employed to verify spatial correlation between the dependent and independent variables and its significant patterns.

Results:

- Figura 1 doesn't has scale and north symbol.

- All figures have no scale.

Discussion:

- There is a limitation related to the fact that some lung cancer cases were not diagnosed being only registered on the death certificate. Thus, this incidence was not detected by the Integrator of Hospital Cancer Records.

Reviewer #2: Comments for Plos One - Manuscript entitled 'Spatial assessment of advanced stage diagnosis and lung cancer mortality in Brazil' (PONE-D-21-20112)

In their article, Lima et al provided relevant and innovative information regarding the spatial distribution of advanced-stage at diagnosis and mortality of lung cancer patients and its correlation with the offer of health services and socioeconomic indicators in Brazil. However, the manuscript has significant limitations that will be addressed below:

Introduction

1. Information regarding lung cancer mortality in Brazil is not updated. More recent data are available at: <https: app="" mortalidade="" www.inca.gov.br="">.

2. The study reports that the 5-year survival rate for lung cancer around the world varies between 10 and 20% but do not give a reference for this information. I would like to suggest including data from this paper: Allemani et al. Global surveillance of trends in cancer survival 2000-14 (CONCORD-3): analysis of individual records for 37 513 025 patients diagnosed with one of 18 cancers from 322 population-based registries in 71 countries. Lancet. 2018 Mar 17;391(10125):1023-1075. doi: 10.1016/S0140-6736(17)33326-3.

3. I would like to suggest rewriting the paragraph beginning with 'In countries with a high Human Development Index (HDI), there are high incidence and mortality rates for lung cancer [1] and an elevated proportion of advanced-stage diagnosis – especially in low-income populations that live in regions with a lower supply of health services for primary prevention, diagnosis, and timely treatment [3,5] [...].' The information provided is not clear. Countries with high HDI normally do not include low-income countries as suggested in the text.

Methods

4. The paper should include more information about definitions of the independent variables such as 'income' (per capita income?), 'health plan coverage' (private health insurance coverage?), 'density of diagnostic equipment' (what kind of equipment?), and density of specialist doctors (which specialists are included?).

What is the difference between 'density of general practitioners' and 'density of family health teams'? In case they are different, is there collinearity between these two independent variables?

5. The study should explain briefly what is the 'Integrator of Hospital Cancer Records'. The paper should also include in the paragraph about limitations of the study, what kind of limitations this database presents (e.g., selection bias?, variation about the quality of information around the country?).

6. The ICD-10 codes C33-34 include 'malignant neoplasm of trachea, bronchus or lung' not only 'malignant neoplasms of lung' as reported in the paper.

7. The study should provide information about the numbers of cases (%) excluded (carcinoma in situ, cases not staged, and without information about place of residence and age). This information is essential because they give readers an idea about data quality.

8. I would like to suggest the revision of some acronyms included in the text as there is a correspondence in English for some of them (e.g. 'PNUD' - in English it is 'UNDP' and 'CID-10' - in English it is 'ICD-10') or there is a more comprehensible translation for a foreign reader such as 'National Regulatory Agency for Private Health Insurance and Plans' for 'ANS'.

Results

9. The paper does not inform what the confidence level of the confidence intervals is (95%?). The same problem appears in the Abstract.

10. The abbreviation 'PASD' is used in some parts of the paper 'Proportion of Advanced-Stage Diagnosis'. I suggest not to use it as it is a non-standard abbreviation.

Discussion

11. The first paragraph of this section needs to be rewritten in order to include the main findings of the paper.

References

12. The references style must be reviewed. PLOS uses the reference style outlined by the International Committee of Medical Journal Editors (ICMJE), also referred to as the “Vancouver” style. Example formats are available at https://journals.plos.org/plosone/s/submission-guidelines#loc-references

Figures

13. The quality of all figures (Fig 1-5) is flawed.</https:>

Reviewer #3: This manuscript reports the relation between the proportion of advanced-stage diagnosis of lung cancer and some risk factors and, also, the relation between lung cancer mortality and some risk factors

It was used Moran index to stablish the spatial autocorrelation of each variable and multivariable analysis to stablish the relation between variables. The statistical analysis is not clear in all cases and requires some clarification. The main finding is that there is a high proportion of advanced-stage diagnosis across the Brazilian territory and inequalities in lung cancer mortality, which are correlated with the most developed areas of the country.

The study is of scientific interest but is confusingly presented. I have some questions and suggestions for improvement, as follows:

Specific comments

Abstract:

* The author mentions many time the term correlation, however I suggest using the term association. Correlation is a specific statistical measure that normally not suggest any direction in the association. Which is not the case in this study, where author have a hypothesis of the direction of the association.

* In the sentence ‘The proportion of advanced stage diagnosis was 85.28% (CI 83.3-87.1)’, it does not clear for me what are the denominator. Can the author please clarify?

* In the sentence ‘The multivariate model for the mortality rates was constituted by the variables “Density of facilities licensed in oncology”, “Income”, and “Health plan coverage”.’ Not clear if this are all the variables used in the model or only the significant variables. Also, the author does not explain the objective in use this model. Can the author explain better of each analysis?

Introduction

* In the sentence ‘’Lung cancer staging at diagnosis and its associated mortality are determined by the histological type, socioeconomic conditions of the population, and the availability and quality of health services’. Is this true? Is the diagnosis established based on socioeconomic conditions? Better says that there is a relation or an association, but it seems to me that the diagnosis is not based on socioeconomic conditions of the population.

* I would advise in sentence ‘’…and its correlation with socioeconomic indicators and health service offer.’ Replace correlation per association, already explained above.

Methods

* In the sentence ‘The dependent variables were the proportion…’ and ‘The independent variables were classified in socioeconomic indicators’, better call ‘response variables’ instead of ‘dependent variables’ and ‘explanatories variables’ instead of ‘The independent variables’. The use of the term ‘independent variables’ is not a very good option because what we are trying to find is variables that are not independent of the response variable.

* In the sentence ‘For the correction of the number of deaths, the methodology proposed by Santos and Souza [12] was followed, considering redistribution per sex, age group, completeness of death record, and ill-defined deaths.’ Can the author please summarize this approach in methods?

* In the sentence ‘Cases of carcinoma In Situ (TNM 0) and with no staging indication and residence were excluded.’ What mean TNM 0? Can the authors please clarify?

* In the sentence ‘The crude and adjusted mortality rates (AMR) (per 100,000 inhabitants) were calculated for each IRUA according to the world population [… The 2013 population was employed as a reference and collected from the population estimate: municipality’. The authors adjusted the mortality rates according to the world population and why not adjusted the proportion of advanced-stage diagnosis of lung cancer?

* In the sentence ‘the regions were classified as High-high and Low-low (when the area has surroundings with similar values) and High-low and Low-high (when the values of the surroundings are different).’ Can the authors please define how the reader can interpret the clusters High-high and Low-low?

* In the sentence ‘Multivariate analysis of lung cancer mortality included only the independent variables that presented a statistically significant correlation with the dependent variable.’ Can the authors clarify how this statistically significant correlation was established?

* In the sentence ‘The final model was selected based on the highest values of the likelihood logarithm, and lowest values for the Akaike Information Criterion (AIC) and Schwarz information criterion [20].’ The term ‘likelihood logarithm’ is not vary often used, we used ‘log-likelihood’. Please consider use this term. Also, it better to replace ‘and’ per ‘or’ because we cannot used the 3 at the same time, not always they going in the same direction.

* In the sentence ‘The spatial autocorrelation of residues was assessed after defining the multivariate model, using Moran’s I and the data dispersion histogram.’ Are we using one model with all the variables into the models or are we using different models for each variable? Can the authors clarify?

* Can the authors please clarify what was the multivariable regression models used?

Results

* In the sentence ‘Fig 1. (a) Spatial distribution and (b) spatial autocorrelation of proportion of advanced-stage diagnosis, combined per sex, of lung cancer; (c) spatial distribution and (d) spatial autocorrelation of adjusted mortality rates, for lung cancer, combined per sex, per IRUA, for 2011-2015.’ Please clarified what figures are for women or for men.

* In the sentence ‘Spatial correlation between proportion of advanced-stage diagnosis of lung cancer and (a) Gini index, (b) aging rate, (c) income, and (d) urbanization degree, per IRUA, for 2011-2015.’ Moran Index is to stablish the spatial autocorrelation with a particular variable, so not sure how the author stablishes the spatial relation between two variables, e.g. advanced-stage diagnosis of lung vs Gini index. Can the author claritfy?

* It seems that the authors in figure 2 analysis the data separately by sex, but in figure 3 it seems that authors put everything together

* In the sentence ‘The residues presented normal distribution and Moran’s I was -0.04 (p<0.01)’, it seems that the variables included in the models do not explain all the variability of the data and residuals seems to presente some patterns.

* In the sentence ‘However, due to theoretical plausibility and its capacity to adjust the final model, the variable “Licensed health facilities for cancer treatment” was preserved.’ I tis not very clear what was the process until the final model, and what was the final model. Can the author clarify?

Discussion

* No comments

Reviewer #4: The article deals with a relevant problem for Public Health, in terms of magnitude and mortality – lung cancer. It uses spatial analysis techniques to explore ecological-level relationships between advanced-stage diagnosis and lung cancer mortality (treated as dependent variables in this study) and some socioeconomic indicators and others indicators related to the provision of health services. The units of analysis chosen were the Intermediate Regions of Urban Articulation in Brazil (IRUA). The authors use global and local spatial analysis methodologies as well as multivariate analysis to examine relationships of interest.

The study is relevant, interesting and treats the literature in an adequate and coherent way. However, some points can be improved, and this review has as its main foundation a perspective of work improvement.

The Complete Review was included in an attached document.

6. PLOS authors have the option to publish the peer review history of their article (what does this mean?). If published, this will include your full peer review and any attached files.

Reviewer #1: **Yes: **Adeylson Guimarães Ribeiro

Reviewer #2: No

Reviewer #3: No

Reviewer #4: **Yes: **Alessandra Cristina Guedes Pellini

---

## [Author Response · Author response to Decision Letter 0]

19 Oct 2021

Journal Requirements:

The requirements were met.

2. Thank you for stating the following in the Acknowledgments/ Funding Section of your manuscript: This study was financed in part by the Coordenação de Aperfeiçoamento de Pessoal de Nível Superior – Brasil (CAPES) – Finance Code 001”.

Please remove any funding-related text from the manuscript and let us know how you would like to update your Funding Statement. Currently, your Funding Statement reads as follows: This study was financed in part by the Coordenação de Aperfeiçoamento de Pessoal de Nível Superior – Brasil (CAPES) – Finance Code 001”. The funders had no role in study design, data collection and analysis, decision to publish, or preparation of the manuscript.

The declaration does not need to be changed. It was removed from the manuscript.

The below paragraph has been added to the method:

This study used secondary data collected from health information systems, which are open and freely accessible. These systems do not provide individual identification data and therefore the approval by a Research Ethics Committee (CEP) was not required, following Resolution 580/2018 [29]. 

4. We note that Figures 1, 2, 3, 4, 5, S1 & S2 in your submission contain map images which may be copyrighted. 

The maps present in the manuscript were prepared by the authors, from a publicly available geographic mesh (shape files), and free of copyright, by the Brazilian Institute of Geography and Biostatistics (IBGE). The geographic meshes are made available without any associated epidemiological data. All maps related to advanced-stage diagnosis and lung cancer mortality by IRUA were prepared by the authors.

The territorial meshes can be found on the link: https://www.ibge.gov.br/geociencias/organizacao-do-territorio/malhas-territoriais/15774-malhas.html?=&t=o-que-e

 Information suggested to improve the description of the method, results and discussion of the study was included in the revised manuscript. All relevant data is within the manuscript and its supporting information files.

Reviewer #1:

Methods: 

- Why the educational level was not used as an independent variable in the correlation analysis and multivariate model? It's an appropriate indicator of socioeconomic position. In the fourteen paragraph of the discussion section the authors stated that: "socioeconomic conditions, specially income and education, are crucial factors determining the lines and mortality of lung cancer".

In Brazil, the distribution of income follows the level of education of the population. Regions with higher income have higher levels of education than other regions of the country, as well as a lower illiteracy rate.

For the present study, per capita income and illiteracy rate were first included. The analysis showed high collinearity (-0.888 p < 0.01) between variables. Based on this result and on the need to reduce the number of variables and maps in the article, the authors decided to keep only the per capita income variable in further analyses.

- About the Integrator of Hospital Cancer Records. What is the national coverage? Only lung cancer cases from public health services or covers cases from the private system? Are all units that receive cancer patients included in the integrator?

The paragraph has been added to the method:

For the study period, the Integrator consolidates information from 273 hospital information units installed in general or specialized cancer hospitals of public, private or philanthropic origin [14]. The coverage of RHC data is over 70% in the South and North, 68% in the Southeast, 62% in the Northeast and 50% in the Midwest [15].

- How the variables were grouped within the IRUA? By the average? Are the municipal boundaries coincident with the IRUA boundaries?

All data included were collected per municipality and then aggregated to an IRUA territory level by means of averaging.

IRUA respects municipal limits, however, they may not coincide with UF limits. The paragraph below was inserted into the manuscript:

 IRUA concentrate a set of municipalities in the provision of highly complex goods and services, including health services [10]. In this regional division, the territory is organized into metropolises, regional capitals and smaller urban centers, according to municipal boundaries. The distribution of public and private services (such as healthcare, education, security, among others) is considered along with the mobility of the population in search of these services and the regions of influence of the cities, without necessarily following the limits of the FU [11]. 

- The authors used a bivariate correlation analysis. In the statistical analysis I suggest to add the information that Global Moran’s Index and the Local Indicator of Spatial Association (LISA) were employed to verify spatial correlation between the dependent and independent variables and its significant patterns.

The adjustment was made as per the paragraph below:

The Global Moran Index was used to verify the spatial autocorrelation of proportions of advanced-stage diagnosis and adjusted lung cancer mortality rates. The presence of spatial clusters was analyzed using the Local Indicator of Spatial Association (LISA). 

- Figure 1 doesn't has scale and north symbol. All figures have no scale.

The graphic scale and the North symbol have been inserted in all figures.

- There is a limitation related to the fact that some lung cancer cases were not diagnosed being only registered on the death certificate. Thus, this incidence was not detected by the Integrator of Hospital Cancer Records.

The limitation was inserted in the manuscript in the last paragraph of the discussion.

Reviewer #2:

1. Information regarding lung cancer mortality in Brazil is not updated. More recent data are available at: 

Data were updated as requested:

In Brazil, in 2019, there were 16,661 deaths due to lung cancer in men and 12,593 deaths in women, which correspond to an estimated risk of 16.19/100,000 men and 9,84/100,000 women [2].

2. The study reports that the 5-year survival rate for lung cancer around the world varies between 10 and 20% but do not give a reference for this information. I would like to suggest including data from this paper: Allemani et al. Global surveillance of trends in cancer survival 2000-14 (CONCORD-3): analysis of individual records for 37 513 025 patients diagnosed with one of 18 cancers from 322 population-based registries in 71 countries. Lancet. 2018 Mar 17;391(10125):1023-1075. doi: 10.1016/S0140-6736(17)33326-3.

The reference was included in the manuscript:

In most countries, the survival of patients with lung cancer at 5 years after diagnosis is only 10% to 20% [4].

4. Allemani C, Matsuda T, Di Carlo V, Harewood R, Matz M, Nikšić M, et al. Global surveillance of trends in cancer survival 2000-14 (CONCORD-3): analysis of individual records for 37 513 025 patients diagnosed with one of 18 cancers from 322 population-based registries in 71 countries. Lancet. 2018; 391:1023-1075. doi: 10.1016/S0140-6736(17)33326-3.

3. I would like to suggest rewriting the paragraph beginning with 'In countries with a high Human Development Index (HDI), there are high incidence and mortality rates for lung cancer [1] and an elevated proportion of advanced-stage diagnosis – especially in low-income populations that live in regions with a lower supply of health services for primary prevention, diagnosis, and timely treatment [3,5] [...].' The information provided is not clear. Countries with high HDI normally do not include low-income countries as suggested in the text.

The information considers that even in countries with a high HDI, there is an unequal income distribution. In this context, the populations of these countries that live on lower incomes live in areas with a reduced supply of health services. This reality contributes to advanced diagnosis and mortality from lung cancer.

Methods

4. The paper should include more information about definitions of the independent variables such as 'income' (per capita income?), 'health plan coverage' (private health insurance coverage?), 'density of diagnostic equipment' (what kind of equipment?), and density of specialist doctors (which specialists are included?).

What is the difference between 'density of general practitioners' and 'density of family health teams'? In case they are different, is there collinearity between these two independent variables?

A table describing each variable was inserted in the manuscript.

The density of general practitioners involves professionals linked to the public and private health system. On the other hand, the density of family health teams partly indicates the distribution of the number of doctors, nurses, nursing technicians and community health agents in the territory linked to the Unified Health System (SUS). In this type of reorganization of primary care, the teams are responsible for a reference territory, in which they accompany registered users and actively search for compulsory notification diseases and injuries, among other activities (BRASIL, 2012). 

Brasil. Ministério da Saúde. Secretaria de Atenção à Saúde. Departamento de Atenção Básica. Política Nacional de Atenção Básica / Ministério da Saúde. Secretaria de Atenção à Saúde. Departamento de Atenção Básica. – Brasília : Ministério da Saúde; 2012.

The collinearity test between the variables resulted in a coefficient of 0.020 and a p-value of 0.80, therefore, they do not present collinearity.

5. The study should explain briefly what is the 'Integrator of Hospital Cancer Records'. The paper should also include in the paragraph about limitations of the study, what kind of limitations this database presents (e.g., selection bias?, variation about the quality of information around the country?).

The paragraph has been added to the text:

For the study period, the Integrator consolidates information from 273 hospital information units installed in general or specialized cancer hospitals of public, private or philanthropic origin [14]. The coverage of RHC data is over 70% in the South and North, 68% in the Southeast, 62% in the Northeast and 50% in the Midwest [15].

The limitations of the study are present in the last paragraph of the discussion.

6. The ICD-10 codes C33-34 include 'malignant neoplasm of trachea, bronchus or lung' not only 'malignant neoplasms of lung' as reported in the paper.

The explanatory text below has been inserted into the method.

The expression tracheal, lung, and bronchial cancer was reduced to lung cancer to maintain the fluidity and objectivity of the text.

7. The study should provide information about the numbers of cases (%) excluded (carcinoma in situ, cases not staged, and without information about place of residence and age). This information is essential because they give readers an idea about data quality.

A flowchart with information about cases selection has been added to the article.

Fig 1. Flowchart of the selection procedure of lung cancer cases between 2011 and 2015 in the RHC Integrator.

8. I would like to suggest the revision of some acronyms included in the text as there is a correspondence in English for some of them (e.g. 'PNUD' - in English it is 'UNDP' and 'CID-10' - in English it is 'ICD-10') or there is a more comprehensible translation for a foreign reader such as 'National Regulatory Agency for Private Health Insurance and Plans' for 'ANS'.

The information suggested has been added to the new and revised version of the manuscript.

Results

9. The paper does not inform what the confidence level of the confidence intervals is (95%?). The same problem appears in the Abstract.

The information suggested has been added to the revised version of the manuscript.

10. The abbreviation 'PASD' is used in some parts of the paper 'Proportion of Advanced-Stage Diagnosis'. I suggest not to use it as it is a non-standard abbreviation.

The abbreviation 'PASD' was removed from the text and replaced by the expression 'Proportion of Advanced-Stage Diagnosis'. 

Discussion 

11. The first paragraph of this section needs to be rewritten in order to include the main findings of the paper.

The paragraph was rewritten, but the information was divided based on the studied outcome to avoid a very long paragraph:

“The study showed a weak spatial dependence for the proportion of advanced stage diagnosis in Brazilian IRUAS. There are three IRUAs in the Northeast, two in the North and one in the South and Southeast regions with high proportion of advanced-stage lung cancer diagnoses and one low-low cluster of proportion of advanced-stage lung cancer diagnoses in the North region. Only demographic and socioeconomic contextual variables were correlated with this outcome.

Nevertheless, the adjusted mortality rates for lung cancer were more influenced by social contexts and the offer of health services, leading to the formation of high mortality clusters in IRUA of the Midwest and South and low mortality clusters in IRUA of the North and Northeast regions.”

References

12. The references style must be reviewed. PLOS uses the reference style outlined by the International Committee of Medical Journal Editors (ICMJE), also referred to as the “Vancouver” style. Example formats are available at https://journals.plos.org/plosone/s/submission-guidelines#loc-references

References have been revised and corrected.

Figures

13. The quality of all figures (Fig 1-5) is flawed.

All figures were verified and converted by PACE to formats accepted by the journal and made available in high quality.

Reviewer #3:

Abstract:

* The author mentions many time the term correlation, however I suggest using the term association. Correlation is a specific statistical measure that normally not suggest any direction in the association. Which is not the case in this study, where author have a hypothesis of the direction of the association.

The term association is used with qualitative variables. When variables are quantitative, as in the case of this study, the term correlation is the most appropriate. When testing the correlation, the coefficient generated quantifies the strength and direction of the relationship between two numerical variables, with results ranging from −1 to 1, which indicate a positive or negative correlation. Thus, when considering the type of variables included in the study, the term correlation is the most recommended.

* In the sentence ‘The proportion of advanced stage diagnosis was 85.28% (CI 83.3-87.1)’, it does not clear for me what are the denominator. Can the author please clarify?

The proportion of advanced stage diagnosis was established through the operation:

Cases diagnosed in advanced stage (TNM III and IV) of lung cancer x 100

All diagnosed cases of lung cancer

* In the sentence ‘The multivariate model for the mortality rates was constituted by the variables “Density of facilities licensed in oncology”, “Income”, and “Health plan coverage”.’ Not clear if this are all the variables used in the model or only the significant variables. Also, the author does not explain the objective in use this model. Can the author explain better of each analysis?

All variables that showed a statistically significant correlation with lung cancer mortality were analyzed in different models. The final multivariate model was constituted by the variables "Density of facilities licensed in oncology", "Per capita Income", and "Health plan coverage". This model was selected based on value of the log-likelihood, lowest values of the Akaike Information Criterion (AIC ) and Schwarz information criterion and on theoretical plausibility.

The paragraph below has been inserted into the text:

The multivariate analysis of lung cancer mortality included the explanatory variables that showed a statistically significant correlation with the response variable and non-colinear variables (correlation<0.7). The spatial error, classic, and spatial log regression models were compared (S1 table), and the global spatial effect model selected for the study was the spatial error model. The final model was selected based on the highest value of the log-likelihood, lowest values of the Akaike Information Criterion (AIC) and Schwarz information criterion [28], and on theoretical plausibility.

Introduction

* In the sentence ‘’Lung cancer staging at diagnosis and its associated mortality are determined by the histological type, socioeconomic conditions of the population, and the availability and quality of health services’. Is this true? Is the diagnosis established based on socioeconomic conditions? Better says that there is a relation or an association, but it seems to me that the diagnosis is not based on socioeconomic conditions of the population.

The information has been corrected in the manuscript.

* I would advise in sentence ‘’…and its correlation with socioeconomic indicators and health service offer.’ Replace correlation per association, already explained above.

As this is a study with quantitative variables, the term correlation is the most indicated in the objective.

Methods

* In the sentence ‘The dependent variables were the proportion…’ and ‘The independent variables were classified in socioeconomic indicators’, better call ‘response variables’ instead of ‘dependent variables’ and ‘explanatories variables’ instead of ‘The independent variables’. The use of the term ‘independent variables’ is not a very good option because what we are trying to find is variables that are not independent of the response variable.

The terms have been corrected as suggested.

* In the sentence ‘For the correction of the number of deaths, the methodology proposed by Santos and Souza [12] was followed, considering redistribution per sex, age group, completeness of death record, and ill-defined deaths.’ Can the author please summarize this approach in methods?

A descriptive paragraph was inserted into the method:

For the correction of the number of deaths, the methodology proposed by Santos and Souza [18] was followed, considering redistribution per sex, age group, completeness of death record, and ill-defined deaths. For the correction of the number of deaths, firstly a correction factor was calculated for the analyzed period, considering age group, sex, and state, from the percentage difference between the number of deaths reported to SIM and the number of deaths redistributed based on chapter II (neoplasms) of the ICD-10. The value 1 was added to the difference obtained to obtain the correction factor per state, as SIM only provides redistributed data per FU. Finally, the correction factor is multiplied by the number of deaths in each municipality, across all Brazilian states [18].

* In the sentence ‘Cases of carcinoma In Situ (TNM 0) and with no staging indication and residence were excluded.’ What mean TNM 0? Can the authors please clarify?

TNM 0 corresponds to stage 0 of the disease. In this case, the primary tumor is considered carcinoma in situ, with no regional and distant lymph node metastases.

* In the sentence ‘The crude and adjusted mortality rates (AMR) (per 100,000 inhabitants) were calculated for each IRUA according to the world population [… The 2013 population was employed as a reference and collected from the population estimate: municipality’. The authors adjusted the mortality rates according to the world population and why not adjusted the proportion of advanced-stage diagnosis of lung cancer?

The proportion of advanced-stage diagnosis was calculated based on hospital records available in the RHC Integrator. Therefore, this is a hospital-based study and not a population-based one, and for this reason, no adjustment was made for the world population.

* In the sentence ‘the regions were classified as High-high and Low-low (when the area has surroundings with similar values) and High-low and Low-high (when the values of the surroundings are different).’ Can the authors please define how the reader can interpret the clusters High-high and Low-low?

The paragraph below has been inserted into the text:

According to the LISA significance level, in the spatial auto-correlation analysis the regions were classified as high-high when the area is formed by IRUA with a high frequency of the variable and also surrounded by high-frequency IRUAs; low-low when the area is formed by IRUA with low frequency variable and also surrounded by low frequency IRUAs; high-low when IRUA with high frequency of the variable is surrounded by regions of low frequency, and low-high when IRUA with low frequency of the variable is surrounded by high frequency IRUAs.

* In the sentence ‘Multivariate analysis of lung cancer mortality included only the independent variables that presented a statistically significant correlation with the dependent variable.’ Can the authors clarify how this statistically significant correlation was established?

The paragraph below has been inserted into the text:

LISA bivariate analysis was performed within the GeoDa 1.6.61 software [27] to assess the spatial correlation between the outcomes studied (proportion of cases diagnosed at an advanced stage and adjusted mortality rate for lung cancer) and each explanatory variables. The analysis generated the Moran Local Index, maps and scatter plot of correlations. The validation of Moran’s I was carried out by a random permutation test, with 99 permutations [27]. Analysis of the surroundings used a first-order Queen Contiguity criterion.

* In the sentence ‘The final model was selected based on the highest values of the likelihood logarithm, and lowest values for the Akaike Information Criterion (AIC) and Schwarz information criterion [20].’ The term ‘likelihood logarithm’ is not vary often used, we used ‘log-likelihood’. Please consider use this term. Also, it better to replace ‘and’ per ‘or’ because we cannot used the 3 at the same time, not always they going in the same direction.

The adjustment was made in the text.

Regarding the definition of the multivariate model, the three recommended criteria for comparison were used, the log-likelihood, Akaike Information Criterion (AIC) and Schwarz information criterion. These criteria are analyzed in different ways. Higher log-likelihood values indicate better-adjust models. When choosing alternative models, the best will be the one with the lowest AIC and Schwarz information criterion. In this study, the authors chose to analyze the three criteria for selecting the final multivariate model.

* In the sentence ‘The spatial autocorrelation of residues was assessed after defining the multivariate model, using Moran’s I and the data dispersion histogram.’ Are we using one model with all the variables into the models or are we using different models for each variable? Can the authors clarify?

The analyzed residues correspond to the set of variables present in the final multivariate model

* Can the authors please clarify what was the multivariable regression models used?

The regression models analyzed were spatial error, classic and spatial record and are presented in a supplementary document. This information has been entered into the method.

Results

* In the sentence ‘Fig 1. (a) Spatial distribution and (b) spatial autocorrelation of proportion of advanced-stage diagnosis, combined per sex, of lung cancer; (c) spatial distribution and (d) spatial autocorrelation of adjusted mortality rates, for lung cancer, combined per sex, per IRUA, for 2011-2015.’ Please clarified what figures are for women or for men.

The analysis was performed with combined data for women e men. The figures show the results for both sexes.

* In the sentence ‘Spatial correlation between proportion of advanced-stage diagnosis of lung cancer and (a) Gini index, (b) aging rate, (c) income, and (d) urbanization degree, per IRUA, for 2011-2015.’ Moran Index is to stablish the spatial autocorrelation with a particular variable, so not sure how the author stablishes the spatial relation between two variables, e.g. advanced-stage diagnosis of lung vs Gini index. Can the author claritfy?

In spatial analysis, the Global Moran Index is used to estimate the autocorrelation of a variable. Once the existence of spatial dependence is confirmed, the presence of spatial clusters (clusters) is verified, based on the Local Indicators of Spatial Association (LISA). To assess the spatial correlation between a response variable and each of the explanatory variables, the LISA bivariate analysis is performed. This analysis generates the Local Moral Index, spatial correlation maps (LISA), and statistical significance.

For a better understanding of the procedures adopted in the method, the paragraph below was inserted in the article:

LISA bivariate analysis was performed within the GeoDa 1.6.61 software [27] to assess the spatial correlation between the outcomes studied (proportion of cases diagnosed at an advanced stage and adjusted mortality rate for lung cancer) and each explanatory variables. The analysis generated the Moran Local index, maps and scatter plot of correlations.

* It seems that the authors in figure 2 analysis the data separately by sex, but in figure 3 it seems that authors put everything together

All figures show the result of the combined analysis by sex. The expression “for both sexes” was added in the title of all figures.

* In the sentence ‘The residues presented normal distribution and Moran’s I was -0.04 (p<0.01)’, it seems that the variables included in the models do not explain all the variability of the data and residuals seems to presente some patterns.

The significance of the autocorrelation of residues was corrected in the manuscript and in the supplementary document, the correct value is p = 0.29. Residues were also analyzed for their correlation with lung cancer mortality rate (Moran’s I 0.06; p= 0.12). The data showed that the model's residuals are independent and not correlated with the dependent variable. 

* In the sentence ‘However, due to theoretical plausibility and its capacity to adjust the final model, the variable “Licensed health facilities for cancer treatment” was preserved.’ I tis not very clear what was the process until the final model, and what was the final model. Can the author clarify?

All explanatory variables that showed a statistically significant correlation with the adjusted mortality rates for lung cancer and that did not show collinearity (correlation <0.7) were analyzed to prepare the multivariate model. The different groups of variables were tested using spatial error, classical and spatial log regression models. Based on log-likelihood values, Akaike Information Criterion (AIC) and Schwarz information criterion, the most adjusted and explanatory multivariate model was the spatial error model, which contained the variables per capita income, density of licensed oncology health facilities and health plan coverage. This model was used in the study.

Reviewer #4:

1. Dependent and Independent Variables: 

 The authors bring together some independent variables of the study into a group called “socioeconomic indicators”. However, the indicators that compose it are, in part, demographic (for example: index of aging and degree of urbanization) and, in part, socioeconomic, for example: Gini Index, Income). Thus, I suggest using the term “demographic and socioeconomic indicators”, instead of just “socioeconomic indicators”, changing every time they appear in the text. 

The suggestion has been inserted into the text.

2. Standardization of terms:

 It is worth clarifying that the term “correlation” is being referred to in the text as an autocorrelation or spatial dependence.

The term autocorrelation was used to refer to the spatial dependence of the variables proportion of diagnosis in advanced stage of lung cancer and adjusted mortality rates for lung cancer. The term correlation refers to the bivariate LISA analysis performed between the response variables (proportion of cases diagnosed at an advanced stage and adjusted mortality rate for lung cancer) and the explanatory variables included in the study.

Standardize the use of the term – “neoplasm” or “cancer” – throughout the text. The neoplasm can be malignant or benign, and it is called “cancer” when it is malignant. I therefore suggest always using the word “cancer”.

The term has been standardized in the text.

3. Figures: 

 Some maps lack essential cartographic elements, such as North arrow (Figures 1, S1 and S2), scale bar (all Figures – 1 to 5, S1 and S2); description of the spatial unit of analysis in the legend (all Figures).

North symbol and graphic scale have been added to all figures. The spatial analysis unit was the IRUA, this description is in the titles of all figures and was included in the legend of the maps.

 The standardization of colors in Fig. 1, in Moran Lisa Local maps, is different from the others. As it is a qualitative variable, I suggest following the same pattern for all of them, with the color pattern in Figures 2 to 5 discriminating better.

The standardization of colors in Fig 1 (in the new document, Fig 2) has been corrected. The variables analyzed in the Moran Lisa Local maps are quantitative. The authors chose to keep the colors of the maps as they correspond to the results of the autocorrelation of the response variables. Furthermore, it facilitates comparison with spatial distribution maps. The other figures show the results of the correlation between the response variables and the explanatory variables.

Also take care of the shade of colors, for example, in Figures 3, 4 and 5 the shade of red (High-High category) of the legends is different from what is observed on maps. 

The standardization of colors has been corrected.

 In the graphics that accompany the maps, I suggest increasing the font size of the X-axis labels and Moran's I values above the graphics. Also pay attention to the correct rounding of the Moran values presented below the maps, with only two decimal places, which must be in accordance with the values presented above the graphs.

Suggestions were met in the revised manuscript. Regarding the Moran’s I values above the graphs, the authors chose to remove them to avoid the repetition of information.

 It is always interesting to describe on the map what each one is, it can be in the upper right corner – for example, Gini index, Aging rate, Income, etc. 

 Specifically in Fig. 3, the Y axis of the last graph has a different title from the others. 

 Also improve the axes titles of the residual analysis graph (Fig. S2). 

Suggestions were met in the revised manuscript.

 In Fig. 1, review the choice of criteria for the categories in the legend in Fig. 1a, as it has little discrimination. The first category ranges from 0 to 81, and the others range from 81 to 100. That is, in the first group there are units with proportions from 0 to 80... which does not help to discriminate the areas on the map, and hinders the discussion. I suggest redoing this categorization. For example, in the far west of the North region, the impression it gives is that it is spared, but in fact it is because it covers values between 0 and 80. Then it becomes more difficult to discuss, especially when, in the Discussion (Paragraph #5), the authors say that “In this study, it was observed that the diagnosis of lung cancer at an advanced stage is prevalent in almost the entire Brazilian territory”. 

The categorization of the maps of the spatial distribution of the proportion of advanced-stage diagnosis and adjusted lung cancer mortality rates was modified for equal intervals.

 The criteria for choosing the categories of quantitative maps (Proportion and Mortality) must be included in the Methods Section – eg. quantiles, equal intervals, natural breaks, etc. 

The criterion used to define the categories of quantitative maps was equal intervals. The information was inserted in the manuscript.

 In all maps, include the layer of the Regions of Brazil, in order to help in the discussion of the findings.

It was not possible to include the Brazil Regions layer on the maps. Due to the amount of information presented in the analysis maps, it was decided to add the maps with the five regions of Brazil and with the 161 intermediate regions of urban articulation as Supporting information. The file can make the manuscript discussion easier to understand.

The map with the five regions of Brazil was prepared by the authors, based on a publicly available geographic mesh (shape files), and free of copyright, from the Brazilian Institute of Geography and Biostatistics (IBGE). The territorial mesh can be found at the link: https://metadados.inde.gov.br/geonetwork/srv/api/records/046a1be6-7c41-427d-8d41-32e47345df80

 In the titles of all Figures, after “per IRUA”, add “Brazil, 2011-2015.”

Suggestions were met in the revised manuscript.

 In the titles of Fig. 1, where it refers to “combined per sex”, substitute “for both sexes”

Suggestions were met in the revised manuscript.

1. Summary: 

 Describe the “Objective” exactly as it appears in the body of the text, at the end of the Introduction Section.

The purpose of the abstract was corrected in the revised manuscript.

 When presenting the results of the multiple analysis, describe how the indicators were related to the outcome (standardized mortality), and whether it was positive or negative and significant or not. 

The information has been inserted in the summary:

The per capita income presented positive correlation and health plan coverage negative correlation with lung cancer mortality rates. Both correlations were statistically significant. The variable Density of Licensed Oncology Health Facilities presented positive correlation with the studied outcome, although without statistical significance. 

Also describe the results of autocorrelation for Mortality (which socioeconomic and demographic, and health care offer variables were significantly correlated), as well as was this was done for the outcome of Proportion of diagnosis in advanced stage.

Due to the high number of variables analyzed (10) and to avoid the repetition of information in the text, it was decided to cite the results of the correlations in a generalized manner and describe only the result of the variable that did not present a statistically significant correlation:

The adjusted mortality rates for lung cancer were a positive and statistically significant correlation with all the demographic and socioeconomic and health service offer indicators, except for the "density of family health teams" (Moran's I -0.01 p = 0.28).

2. Introduction: Paragraph #1: the authors describe that “lung cancer is the second most incident type of cancer worldwide”. - What is the first type of cancer? Include in text.

The information has been included in the text: 

Global statistics indicate that lung cancer is the second most common type of cancer in the world, following breast cancer, with 2.2 million new cases and 1.8 million deaths estimated for 2020.

 Paragraph #4: the authors refer, at the end, that “The importance of each factor, however, varies with the different disease scenarios for lung cancer”. - Make it clearer what these different disease scenarios are.

The text has been changed to:

“The importance of each factor, however, is highly dependent on regional contexts [5,6].”

These contexts are presented in the next paragraph.

3. Methods:

Paragraph #1: justify the choice of “Intermediate Regions of Urban Articulation (IRUA)”, as the spatial analysis unit.

The information has been included in the text: 

The IRUA constitute a territorial mesh defined based on common characteristics and relationships between municipalities, based on an urban center. This division reflects the context of regions and urban articulations established in the period studied [10], which cannot be portrayed using the delimitation of each individual municipality only. For this reason, the IRUA were chosen as the unit of analysis in this study. Access to this digital territorial mesh is public and freely available at the IBGE website [12].

Paragraph #3: the authors state that: “The IRUA are a type of Urban-Regional Division, which organize the territory into metropoles, regional capitals, and smaller urban centers, considering the flows of public and private management and the existence of more complex urban functions, without necessarily following the limits of FU”. - In relation specifically to “public and private management flows”, what kind of sectors do the authors refer to? Health only or others? Clarify.

The information has been included in the text: 

IRUA concentrate a set of municipalities in the provision of highly complex goods and services, including health services [10]. In this regional division, the territory is organized into metropolises, regional capitals and smaller urban centers, according to municipal boundaries. The distribution of public and private services (such as healthcare, education, security, among others) is considered along with the mobility of the population in search of these services and the regions of influence of the cities, without necessarily following the limits of the FU [11].

Paragraph #3: specify here, too, whether the IRUA boundaries coincide with the UF and municipal boundaries. Given that data (dependent and independent variables) were obtained from municipalities, explain how these data were converted or transformed for the IRUA spatial analysis units. (This is commented a little at the end of the seventh paragraph of Methods, but it would be interesting to bring it here and complement). Do these units of analysis make up a digital territorial mesh available at IBGE? Clarify in the text.

Information about the IRUAs and the availability of territorial geographic mesh (shape files) was inserted in paragraphs 3 and 4 of the study design.

The information has been included in the text: 

All data included here were collected by municipality and then aggregated to an IRUA territory level by means of averaging 

Paragraph #4: “The dependent variables were the proportion of advanced-stage diagnosis and the adjusted mortality rates for lung cancer, combined for men and women, per IRUA, for 2011-2015.” -specify how the adjustment of the mortality rate was made. By gender and age group?

Mortality rates were adjusted by age group for men and women, separately. To analyze the spatial dependence and correlations with the explanatory variables, the age-adjusted rate, combined for both sexes, was used.

Paragraphs #6 and #7: these paragraphs discuss the information sources of the dependent variables; therefore, they should come before the fifth paragraph (which deals with independent variables).

The sequence of paragraphs has been corrected. 

Paragraph #7: presents the ICD-10 C33-C34, however, the ICD-10 C33 is from tracheal cancer. It is important to clarify that whenever the text mentions lung cancer, it is also including tracheal cancer (ICD-10: C33), in addition to bronchial and lung cancers (ICD-10: C34 to C34.9).

An explanatory paragraph about the nomenclature has been added:

The expression tracheal, lung, and bronchial cancer was reduced to lung cancer to maintain the fluidity and objectivity of the text.

Paragraph #8: describe in full what the acronym UNDP means. It also needs to better specify the source of each of the socioeconomic/demographic variables and the variables related to health care. Specify for each variable. If you prefer, put it in a table format, containing each variable, its source, the indicator calculation method and the variable name used in the maps and in the multiple analysis. It was also unclear why the population by sex and age was collected for these variables, since there was no separate analysis of these indicators by sex or age.

The suggested Table was inserted into the manuscript.

The population by sex and age was collected to calculate and present the proportion of advanced-stage diagnosis and adjusted mortality rate from lung cancer for men and women, separately.

Paragraph #8: the authors state that “The calculation of indicators employed as the denominator the population of the 2010 census, per municipality, sex, and age as reported by IBGE.” Why was not used a more recent population that would correspond to the middle of the evaluated period – 2011 to 2015? For example, on the DATASUS website, it is possible to obtain the population estimate for 2012, by age group and municipality – this year would be closer in relation to the other events in the study. See: http://tabnet.datasus.gov.br/cgi/deftohtm.exe?ibge/cnv/popbr.def

The population used was from the middle of the period studied. The information has been included in the text: 

The 2013 population was used as a reference for the calculation of indicators along with data on municipality, sex and age, as published by IBGE [25]. 

Paragraph #9: “Cases of carcinoma In Situ (TNM 0) and with no staging indication and residence were excluded.” - It is important for the authors to discriminate, in the Results, how many cases (N and %) represented carcinoma in situ, how many cases had no staging indication, and how many cases did not contain place of residence. In general, in cancer registries, we have difficulty calculating incidence for smaller locations, due to gaps in case address data. I believe it is not the case with information about the municipality of residence, but it is worth sizing up. This paragraph would look better with those referring to dependent variables. In the Results, it would be important to inform how much the proportions of advanced-stage diagnosis of lung cancer varied between the IRUA (minimum and maximum values and where they occur), as well as how much the mortality rates varied (minimum and maximum and where they occur).

Suggested information was inserted in the method (Flowchart of the selection procedure of lung cancer cases between 2011 and 2015 in the RHC Integrator ) and in the manuscript results.

Paragraph #10: “The 2013 population was employed as a reference and collected from the population estimate: municipality, sex, and age, as reported by IBGE.” - This was confused by the previous statement (in the Paragraph # 8) that “The calculation of indicators employed as the denominator the population of the 2010 census, per municipality, sex, and age as reported by IBGE.” - If this last information refers to the calculation of socioeconomic and demographic indicators, it needs to clarify the reason for this option.

The population used for all calculations was from 2013. The correction was made in the manuscript.

Paragraph #10: The crude and adjusted mortality rates (AMR) (per 100,000 inhabitants) were calculated for each IRUA according to the world population”. - Specify here how the adjustment was made (by sex and age group?), and rewrite so that it is clear that the world standard population was used only for the calculation of the adjusted rates (and not for the crud rates).

Mortality rates were adjusted by age group for men and women, separately. To analyze the spatial dependence and correlations with the explanatory variables, the age-adjusted rate, combined for both sexes, was used.

The paragraph was rewritten:

The crude and adjusted mortality rates (AMR) (per 100,000 inhabitants) were calculated for each IRUA. Age-adjusted rates were calculated from the direct standardization method [23], using the world standard population as reference [24].

Paragraph #11: “Global Moran’s Index and the Local Indicator of Spatial Association (LISA) were employed to verify spatial correlation and its significant patterns, respectively.” - In fact, the Moran Global Index assesses whether there is spatial dependence on the space as a whole, and the Moran LISA assesses these relationships locally, in some locations showing significance and not in others. So, rewrite this sentence, which is confusing (I suggest reviewing the concepts and uses of Moran Global and Local at: http://www.panaftosa.org.br/Comp/MAPA/442913.pdf )

The text was rewritten:

The Global Moran Index was used to verify the spatial autocorrelation of proportions of advanced-stage diagnosis and adjusted lung cancer mortality rates. The presence of spatial clusters was analyzed using the Local Indicator of Spatial Association (LISA).

Paragraph #11: It is important to clarify that the authors use the Moran Local Univariate for each dependent variable, and analyze the Moran Local Bivariate between the independent and dependent variables. This needs to be clarified, as well as in the low-high and high-low patterns, which order is specified – VD-VI or VI-VD. Also specify what the graphics that appear beside each map mean.

The suggested information was inserted in the text:

According to the LISA significance level, in the spatial auto-correlation analysis the regions were classified as high-high when the area is formed by IRUA with a high frequency of the variable and also surrounded by high-frequency IRUAs; low-low when the area is formed by IRUA with low frequency variable and also surrounded by low frequency IRUAs; high-low when IRUA with high frequency of the variable is surrounded by regions of low frequency, and low-high when IRUA with low frequency of the variable is surrounded by high frequency IRUAs.

The suggested information was inserted in the text:

“The analysis generated the Moran Local index, maps and scatter plot of correlations.”

Paragraph #12: why was multiple analysis done only for mortality, but not for the proportion of lung cancer cases with advanced diagnosis?

The moderate spatial dependence of the proportion of diagnoses in advanced stage of lung cancer (Moran’s I = 0.37) and the weak correlations with the explanatory variables studied (as seen in figures 3 and 4) made it impossible to design a multiple analysis model.

Paragraph #12: the last sentence “The final multivariate model included statistically significant variables and those with theoretical plausibility.” - is in disagreement with the first: “Multivariate analysis of lung cancer mortality included only the independent variables that presented a statistically significant correlation with the dependent variable”. It seems to me that only the last one in the paragraph is correct, as the final model kept only one variable with theoretical plausibility. It is also important to say with which models the spatial error model was compared, and to refer to the S1 table.

The adjustment was made to the text (paragraph 5 of the statistical analysis).

4. Results:

Paragraph #1: include information on maximum and minimum variations (and respective locations where they were found) of proportions (PASD) and mortality. When it is said that “The formation of only one low-low cluster located in the North region was observed.”, it would also be important to refer to the other high-high clusters found.

Information about the variations was inserted in the text:

“The IRUAS of Tefé and Tabatinga, located in the North region, had no records of cases of lung cancer diagnosed at an advanced stage (proportion of 0%). The IRUAS of Juazeiro do Norte, Ararapina, Guarabira, Afogados da Ingazeira (located in the Northeast region), Lavras (Southeast region) Parintins (North region), and Cáceres and São Félix do Araguaia (Midwest) presented a proportion of 100% for cases diagnosed in stage III or IV.”

“The Itaberaba and Bom Jesus da Lapa IRUAS, located in the Northeast region, presented the lowest adjusted lung cancer mortality rates, 3.95 and 4.95 deaths per 100,000 inhabitants, respectively. The highest adjusted rates were verified in the IRUAS of Brasília (39.68 deaths per 100,000 inhabitants), in the Midwest region, and Santa Cruz do Sul (29.49 deaths per 100,000 inhabitants), in the South region of Brazil.”

Regarding the formation of clusters of the proportion of advanced stage lung cancer diagnosis, the results show that only a low-low cluster was formed (in the North region). There is no high-high cluster formation, only a few isolated IRUAs showed a high proportion of advanced-stage diagnosis.

 Paragraph #2: specify how many more cases the correction was able to add to the total N of deaths, that is, why is this correction work important?

“After correction, 29,371 deaths from lung cancer were added to the initial number, which corresponds to 19.3% of the deaths included in the study. The number of deaths resulted in 151,699.”

Paragraph #2: where it reads “For the combined AMR for men and women, the average was 12.82 (SD 5.12) deaths per 100,000 inhabitants.”, replace with: “The AMR including men and women was 12.82 (SD 5.12) deaths per 100,000 inhabitants”. As it stands, it looks like an average of the rates for men and women was taken... would that be the case?

The average of the adjusted mortality rates was calculated, combined for men and women, based on the values of the IRUAs.

 Paragraph #3: “The independent socioeconomic variables demonstrated low spatial correlation with advanced -stage diagnosis, with statistical significance for Gini (p=0.01), aging rate (p=0.02), and income (p=0.01)” - also include the values of the Moran’s I. About the statement: “Regarding the variables related to the offer and availability of health services, no statistically significant correlations were observed with proportion of advanced stage diagnosis, as displayed in Fig 3”. - specify that no global spatial dependence was found (as at local levels they were observed, as seen in LISA maps).

Moran Index values were entered.

The global spatial dependence of a variable is verified using the global moran index. In the case mentioned above, the correlation does not generate the global moran index, but the local moral index related to the correlation between the studied variables. The results show that the existing correlations are very weak and without statistical significance.

 Paragraph #4: “Fig 4 and 5 show the correlation between AMR, combined by sex, for lung cancer and the socioeconomic and health service offer indicators. - Replace with: “Fig 4 and 5 show the correlation between AMR, for both sex, for lung cancer and the socioeconomic and health service offer indicators, respectively.” 

The text was adjusted as suggested.

Table 1: Consider revising the title to: “Spatial regression analysis of adjusted lung cancer mortality rates and its correlation whit socioeconomic and health service offer indicators, per IRUA, Brazil, 2011-2015.” At the bottom of the table, there appears to be an error, as what is itemized as “p = 0.663” looks like it would have to be “R2 = 0.663”. 

The text was adjusted as suggested. The "p" has been corrected for R2. 

Paragraph #6: Replace “was preserved” by “was kept in the model.”.

The text was adjusted as suggested.

5. Discussion:

Paragraph #3: “Although there is consensus in the literature on the histological types of lung cancer with a higher risk of diagnosis at an advanced stage, there are still inconsistent results regarding the effect of socioeconomic conditions and the provision of health services on the stage of the neoplasia in the diagnosis” – Better explain what the authors mean by “there are still inconsistent results”, that is, what is the knowledge gap in relation to what is stated, or discuss in accordance with other references. 

Studies have found different results for factors associated with stage of lung cancer diagnosis. In the following paragraph of the manuscript some of these studies are shown. It is possible to notice that there is still no consensus on what are all the factors associated with the outcome, as well as the power of the association. For a better understanding of the idea to be transmitted, it was decided to change the expression “inconsistent results” to “different results”.

Paragraph #5: Replace “PASD were positively correlated with the aging rate and income and inversely correlated with Gini” with “PASD were positively correlated with the aging rate and income and inversely correlated with Gini Index”.

The text was adjusted as suggested.

Paragraph #7: “...encompassing the North and Northeast regions, with the worst socioeconomic índices” – Highlight the findings in Fig. 2 (low-low pattern for the North Region) and Fig. 4 (low-low pattern for the North and Northeast region), with regard to income and urbanization.

The findings were highlighted as suggested.

 Paragraph #8: It is better not to talk about incidence, as this indicator was not evaluated in the study. As for considerations about reducing the smoking habit in Brazil, and its impact on the incidence of lung cancer, it is worth discussing in light of the findings of the Vigitel studies and INCA: https://www.inca.gov.br/observatorio-da-politica-nacional-de-controledo-tabaco/dados-e-numeros-prevalencia-tabagismo

The text on the incidence has been removed.

The discussion on tobacco consumption is included in the analysis of mortality.

Paragraph #8: At the end of the paragraph, it is stated that: “...the proportion of cases at advanced stages can accompany the incidence of lung cancer at the Brazilian IRUAs.” I believe this statement is fragile, given, above all, that the diagnostic capacity and resources are different in different regions of Brazil. 

The text was modified according to the following paragraph:

The association between higher incomes, aging rates, and lower Gini index is verified in territories with higher development indices which, in Brazil, coincide with IRUAs that present better offer of health services and older population [33,36]. The high proportions of lung cancer cases diagnosed at a late stage in more developed regions can reflect a better quality of the registries of cases diagnosed in these territories [37].

 Paragraph #13: Likewise, one should avoid talking about “incidence”, but report the evaluated indicator – “proportions of advanced-stage diagnosis of lung cancer”. 

The text on the incidence has been removed.

 Paragraph #15: “Regarding the offer and availability of health services, in this study, the high mortality rates were correlated with territories that presented low coverage of health plans and high density of facilities licensed in oncology.” - This sentence expresses, in part, the findings of the spatial regression analysis, referring to the “health plan coverage”; but not in relation to the “high density of facilities licensed in oncology”, which was not significant, and was maintained only for conceptual reasons. If analyzed from the point of view of Fig. 5, spatial dependence was positive in the bivariate analysis, in relation to the coverage of health plans, as well as in relation to the high density of oncology services. A possible explanation may be due to the fact that these areas are sought after by people seeking treatment for their cancer, who may even take up residence in them. It is important to discuss these points more clearly, and about the discrepancy between the findings in the different analyzes (Spatial regression analysis and Moran’s I).

The discussion about the change of residence address was inserted into the manuscript (paragraph 22).

Bivariate analysis showed a positive correlation between indicators of health insurance coverage and density of facilities licensed in oncology and the response variable (adjusted mortality rates for lung cancer). But it was possible to see, through the maps, that there are clusters of high mortality in regions with low coverage of health plans, especially in IRUAs in the South region.

In the regression model, the density of facilities licensed in oncology lost significance and the coverage of health plans showed an inverse correlation with lung cancer mortality. This discrepancy in results may be related to the levels of socioeconomic inequalities that exist even in the most developed regions, as discussed in the text. This makes access to public or private health services unequal among populations.

Changes in address, in search of treatment, can also move part of the population with higher income to other regions with other health services, including private ones. Likewise, people who need access to health services through the SUS can migrate to places with a greater offer of public services. Both situations can contribute to the reduction of health plan coverage in regions with high mortality rates.

Paragraph #18: “In the South, the research evidenced IRUA with high mortality rates for lung cancer and low coverage of health plans.” - This is an assertion resulting from the finding expressed in Fig. 5, whose legend indicates that the South region has a low-high standard in relation to the health plan coverage; hence the need to clarify, in the Methods Section, that the first indicator – low – refers to the independent variable (in this case – health plan coverage) and the second – high – refers to mortality rates; that is, make it clearer, so as not to give the opposite impression. It can be clarified in the Methods (that the first word of the pattern refers to VI and the second to VD). 

The explanation was inserted in paragraph 4 of the Statistical analysis of the manuscript section.

Paragraph #20: “...and areas with a more comprehensive offer and availability of health facilities licensed in oncology could result from high number of diagnoses of the disease, more structured surveillance and better quality death records in these places”– adding to this the fact that people can take up residence in these locations in order to facilitate their access to treatment.

The suggestion was inserted in the text, as shown below:

In addition, migration of people to these regions in search of cancer diagnosis and treatment services is highlighted. These populational displacements can concentrate mortality rates in more developed regions [50].

Paragraph #21: “Eight presented higher standards, five were compatible, and 16 presented lower standards than suggested by the Therapeutic Directives.” - How many of these 16 services were exclusively of Supplementary Health? Considering that many private services also provide services to the SUS... Paragraph #22: “These differences emphasize that most treatment centers offer lowerquality care than preconized by the Ministry of Health and provided by Supplementary Health.” – It is not clear whether SUS establishments follow the Ministry of Health Therapeutic Directives less and that Supplementary Health performs better in this regard. Apparently this is what is said, but it must be justified, if possible, on the basis of numbers. In the previous paragraph, this is also not very clear. Please rewiew. 

The study was carried out with 52 cancer centers linked to the Unified Health System (SUS). The data referring to the treatment were compared to what is recommended by the Ministry of Health, through the Diagnosis and Treatment Guidelines, and to the standard practiced in supplementary health, based on the Procedures List of the National Health Agency (ANS).

The text was adjusted to improve the understanding of the paragraph:

The high offer of cancer-related services associated with lung cancer mortality can also evidence the quality of the care offered. The study by Kaliks et al [51] identified the differences in public (SUS) treatment for the four most incident cancers and compared with the Therapeutic Directives established by the Ministry of Health and the assistance provided by Supplementary Health. Of the 52 SUS cancer treatment centers investigated, only 29 followed the directives regarding lung cancer treatment. Eight presented higher standards, five were compatible, and 16 presented lower standards than suggested by the Therapeutic Directives. Compared to the SS, 19 SUS cancer centers presented standards below those recommended for private institutions.

Paragraph #24: “The analysis shows that the spatial distribution of the proportion of advanced stage lung cancer diagnosis in the Brazilian IRUA is not significantly affected by contextual socioeconomic factors and the offer of health services”. – This statement is in disagreement with Fig. 2, which shows a significant bivariate correlation between the indicator “proportion of advanced stage lung cancer diagnosis” and most of the analyzed socioeconomic and demographic factors, except the “Urbanization degree”.

The text has been adjusted to:

The analysis shows that the spatial distribution of the proportion of advanced-stage lung cancer diagnoses in the Brazilian IRUA is weakly related to the demographic and socioeconomic context.

Paragraph #25: “The cases excluded, due to the absence of data, could present a nonrandom spatial distribution, which was not assessed herein” – again, it is important to include how many of these cases were excluded due to lack of data, that is, how much of the sample, in %, they represented.

The information was inserted in the text through a flowchart of case selection.

6. Conclusion:

 Paragraph #1: “... The PASD were weakly correlated to socioeconomic indicators only.” – Consider replacing with: “The PASD were weakly spatially correlated to socioeconomic indicators only.

The adjustment was made in the text.

---

## [Decision Letter · Decision Letter 1]

10 Dec 2021

PONE-D-21-20112R1Spatial assessment of advanced stage diagnosis and lung cancer mortality in BrazilPLOS ONE

Dear Dr. de Souza,

Thank you for submitting your manuscript to PLOS ONE. After careful consideration, we feel that it has merit but does not fully meet PLOS ONE’s publication criteria as it currently stands. Therefore, we invite you to submit a revised version of the manuscript that addresses the points raised during the review process.

Please see my comments at the end of this email for additional comments.

We look forward to receiving your revised manuscript.

Kind regards,

Edison I.O. Vidal, MD, MPH, PhD

Academic Editor

PLOS ONE

Journal Requirements:

Additional Editor Comments:

I commend the authors for the significant improvement related to the new version of their manuscript. However, there are still several issues that must be addressed before it can be reconsidered for publication, as described by the reviewers.

In their first report on this manuscript, reviewer #3 argued that the authors had often misused the term “correlation” in their text. The reviewer claimed that correlation measures do not suggest any direction regarding the relationship between two variables. On the one hand, that explanation is incorrect because correlation coefficients measure the extent to which two variables have a linear relationship and may be positive or negative, thereby indicating the direction of that relationship. On the other hand, the authors responded to the reviewer’s comment by arguing that “The term association is used with qualitative variables. When variables are quantitative, as in the case of this study, the term correlation is the most appropriate. When testing the correlation, the coefficient generated quantifies the strength and direction of the relationship between two numerical variables, with results ranging from −1 to 1, which indicate a positive or negative correlation. Thus, when considering the type of variables included in the study, the term correlation is the most recommended.” Unfortunately, the authors’ argument also has several important flaws. First, it is incorrect to claim that the word “association” should be restricted to the relationship between qualitative variables. Although association and correlation are sometimes loosely used as synonyms, in statistics, correlation is more commonly used to refer to correlation coefficients, whereas association is a more generic term denoting a statistical dependence between two or more variables that may also be positive or negative. Moreover, correlation coefficients reflect the degree of linear association, i.e. the extent to which the points of the scatter plot of two variables are close to a monotonic line but say nothing about the slope of that line, which would be better described the coefficient of a linear regression equation. Hence, for the sake of clarity, I would like to recommend that the authors restrict the use of the term correlation to situations where they refer specifically to correlation coefficients or spatial autocorrelation measures and that they use the term association when referring to the coefficients derived from the equations of bivariate and “multivariate” spatial regression analyses throughout the text. By the way, “multivariable” regression is a more accurate term for the analyses performed in this study than “multivariate” regression. See the following article by Hidalgo and Goodman (https://www.ncbi.nlm.nih.gov/pmc/articles/PMC3518362/)

With regards to the association vs. correlation nomenclature, I recommend checking Miguel Porta’s Dictionary of Epidemiology (https://www.oxfordreference.com/view/10.1093/acref/9780199976720.001.0001/acref-9780199976720). As to the interpretation of correlation coefficients, I would like to suggest the following article by Patrick Schober (https://pubmed.ncbi.nlm.nih.gov/29481436/)

Reviewers' comments:

Reviewer's Responses to Questions

**Comments to the Author**

1. If the authors have adequately addressed your comments raised in a previous round of review and you feel that this manuscript is now acceptable for publication, you may indicate that here to bypass the “Comments to the Author” section, enter your conflict of interest statement in the “Confidential to Editor” section, and submit your "Accept" recommendation.

Reviewer #1: All comments have been addressed

Reviewer #2: All comments have been addressed

Reviewer #3: (No Response)

Reviewer #4: All comments have been addressed

2. Is the manuscript technically sound, and do the data support the conclusions?

Reviewer #1: Yes

Reviewer #2: Yes

Reviewer #3: Yes

Reviewer #4: Yes

3. Has the statistical analysis been performed appropriately and rigorously? 

Reviewer #1: Yes

Reviewer #2: Yes

Reviewer #3: Yes

Reviewer #4: Yes

4. Have the authors made all data underlying the findings in their manuscript fully available?

Reviewer #1: Yes

Reviewer #2: Yes

Reviewer #3: Yes

Reviewer #4: No

5. Is the manuscript presented in an intelligible fashion and written in standard English?

Reviewer #1: Yes

Reviewer #2: No

Reviewer #3: Yes

Reviewer #4: Yes

6. Review Comments to the Author

Reviewer #1: All my comments have been addressed and I consider that the manuscript is suitable for publication in this version.

Reviewer #2: Comments for Plos One - Manuscript entitled 'Spatial assessment of advanced stage diagnosis and lung cancer mortality in Brazil' (PONE-D-21-20112-R1)

In their manuscript, Lima et al. provided relevant and innovative information regarding the spatial distribution of advanced stage at diagnosis and mortality of lung cancer patients and its correlation with the supply of healthcare services and socioeconomic indicators in Brazil. I consider that the new version (R1) of the manuscript improved considerably and all comments were addressed by authors. I have only some small comments regarding the manuscript:

Authors´ affiliation

1. I think there is an error with the affiliation of the first author as in the manuscript appears two different Postgraduate Programs: 'Graduate Program in Collective Health' and 'Graduate Program in Public Health' belonging to the same Department (Public Health or Collective Health?).

Abstract

2. I would like to suggest rewriting the sentence 'The adjusted mortality rates for lung cancer presented positive and statistically significant correlation with all the demographic and socioeconomic and health service offer indicators [...].' to 'The adjusted mortality rates for lung cancer presented a positive and statistically significant correlation with all demographic, socioeconomic and healthcare services supply indicators [...].'

The term 'health service offer' appears many times in the manuscript, I would like to suggest considering using 'healthcare services supply' or 'supply of healthcare services'.

3. I would like to suggest to change the term 'Health plan coverage' to 'private health insurance coverage'. The authors sometimes also use the term 'Supplementary Health', I suggest unifying them.

Introduction

4. I would like to suggest to chance the term 'advanced stage diagnosis' to 'advanced-stage diagnosis' or to 'advanced stage at diagnosis'.

Materials and methods

5. Sometimes, the authors use the term 'Federation Units (FU)' other 'states'. Foreign people do not know if they are the same or not. Please, consider unifying them.

The same problem happens to 'Integrator of Hospital Cancer Records'. Sometimes the authors use 'Integrator' other 'RHC' or 'IRHC' or 'RHC Integrator'. I suggest using the abbreviation 'Integrator-HBCRs' to the term 'Integrator of Hospital-Based Cancer Registries'.

6. Lung cancer is morphologically classified into different subtypes. Because of this, I suggest changing the term 'carcinoma in situ' to 'in situ disease'.

7. Please correct the spelling of 'non-colinear variables' to 'non-collinear variables'.

Results

8. I would like to suggest changing the term 'lung cancer adjusted mortality rates' to 'age-adjusted mortality rates for lung cancer' or to 'age-adjusted lung cancer mortality rates'.

Figure 1

9. I would like to suggest changing the term 'without staging' to 'stage not known'.

10. I would like to suggest changing the term 'carcinoma in situ' to 'in situ disease' or to 'not invasive cancer'.

11. Please, change the sentence '31,010 (73.2%) cases of lung cancer (age between 18 and 99 years were included)' to '31,010 (73.2%) cases of lung cancer (age between 18 and 99 years) were included'.

S2 Fig

12. Include in the title the term 'Brazil'.

S3 Fig

13. Include in the title the term 'Brazil'. Further, include the information about the meaning of x-axis and y-axis.

Table 1

14. Include in the footnote the meaning of the abbreviations used in the table

15. I would like the confirmation of the authors if the description they used for the variable 'adjusted mortality rates for lung cancer' is correct: 'Lung cancer mortality rate, combined for men and women, adjusted for age and standard world population'.

Reviewer #3: In relation to the topic 'correlation' and its use, I was not referring to the measure that varies between -1 and 1 and which is a correlation. The author appears to be using the term correlation when it is a measure of correlation and when it is not. For example, in the title of Figure 2, 'Spatial regression analysis of adjusted lung cancer mortality rates and their correlation with socioeconomic indicators and health service provision, by IRUA, 2011-2015', the author used the term correlation for the coefficients estimated by the regression model, this measure does not vary between -1 and 1, and it is not a correlation. It is more appropriate to use the term association between adjusted mortality and socioeconomic indicator rather than correlation.

Can the author please review?

Reviewer #4: The manuscript showed substantial improvement after the first review; however, I noticed that some aspects can still be improved. The new suggestions that I present, although they are relevant in number, are easy to resolve.

1. Figures:

Fig. 1: change the acronym “RHC”, which is in Portuguese, for “HCR” (Hospital Cancer Records).

S1 table: fix the name of the “spatial lag” model (this is described as “spatial log”).

2. Abstract:

Standardize the number of decimal places. For example: in lines 40 and 41 of the Abstract: “The proportion of advanced-stage diagnosis was 85.28% (95% CI 83.3-87.1).” – The proportion value (85.28%) appears with two places, and the 95% confidence interval with only one (95% CI 83.3-87.1). Standardize (I suggest keeping everything to two decimal places, then getting the CI values right). Also check this in the rest of the text.

Line 43: replace “Gini” by “Gini Index”.

Line 45: replace “significant correlation” by “significant spatial correlation”.

Lines 48 and 52 differ in the denomination of the variable related to Oncology Services – in line 48, it is described as “Density of licensed facilities in oncology”, and in line 52, “Density of Licensed Oncology Health Facilities”. Standardize the term, as well as the rest of the text.

Lines 51-53: replace the sentence: “The variable Density of Licensed Oncology Health Facilities presented positive correlation with the studied outcome, although without statistical significance.” by: “The variable Density of Licensed Oncology Health Facilities showed no significant correlation with lung cancer mortality rate”.

3. Introduction:

Line 62: the term “neoplasm” still appears in the text. Change the term “neoplasms” to “cancers” (including in reference [1] – GLOBOCAN, it is described as “cancers” and not “neoplasms”).

Lines 64-66: “In Brazil, in 2019, there were 16,661 deaths due to lung cancer in men and 12,593 deaths in women, which correspond to an estimated risk of 16.19/100,000 men and 9,84/100,000 women [2]”. First the author presents the numbers of deaths, then he says that this corresponds to estimates of risks (incidences). Review if the data really correspond to the incidence, in this case, do not use the term “which correspond”, but treat the two sentences as independent. Another option would be to present the number of new cancer cases of each sex, and then correlate the respective risks.

Lines 83-84: replace the phrase: “When considering the socioeconomic and distribution differences of health services in Brazilian regions” by: “When considering socioeconomic differences, as well as the distribution of health services in Brazilian regions”.

4. Material and Methods:

I suggest changing the name of the Subsection: “Study design” to “Study design and Spatial analysis units”, since the authors describe, in this item, several aspects related to the analysis units chosen for the study – IRUA.

In the subsection: “Study variables and data sources”, I suggest changing the order of the paragraphs, initially describing everything that refers to the response variables and, later, everything that concerns the exploratory variables. So, the new order would be:

Paragraph #1: “The response variables were the proportion of late-stage diagnosis and adjusted mortality rates ...”

Paragraph #2: “All data included here were collected by municipality and then aggregated to an IRUA territory level by means of averaging ...”

Paragraph #3: “Based on the TNM Classification of malignant tumors, lung cancer cases were classified as advanced stage (TNM III and IV) and early stage (TNM I and II) ...”

Paragraph #4: “The deaths that occurred in Brazil between 2011 and 2015 due to lung câncer (ICD-10, C33-34) [16] were collected from the Mortality Information System (MIS) ...”

Paragraph #5: “For the correction of the number of deaths, the methodology proposed by Santos and Souza [18] was followed ...”

Paragraph #6: “The crude and adjusted mortality rates (AMR) (per 100,000 inhabitants) were calculated for each IRUA ...”

Paragraph #7: “The explanatory variables were classified into demographic and socioeconomic indicators ...”

Paragraph #8: “Demographic and socioeconomic variables for 2010 were obtained from the Brazilian Atlas of Human Development (UNDP) ...” At the end of this last paragraph, keep the sentence already in lines 162-163: “Table 1 presents the study variables and corresponding descriptions ...”

Below the paragraphs of this Subsection, would then come Fig 1 (Flowchart) and Table 1 (Response and exploratory variables).

Line 118: describe the term “lung cancer” in italics.

Lines 122 and 125: change the acronym “RHC”, which is in Portuguese, to “HCR” (Hospital Cancer Records).

Line 129: replace the term “for 2011-2015” by “for each year”, as the way it is is repeated in relation to line 127, where the years were already mentioned.

Line 130: replace the term “residence” by “residence address”.

Lines 131 and 133: the excerpt “For the correction of the number of deaths” was repeated. I believe it can be deleted from line 133.

Line 135: does the word “state” mean “Federated Unit”? If so, it would be better to substitute by this last term.

Lines 135 and 138: replace the acronym “SIM” by “MIS” (as defined this acronym in line 128).

Line 146: the acronym UNPD refers to the Organ: “United Nations Development Programme”, which needs to be properly described in full.

Lines 147 and 148: the abbreviations “CNES” and “ANS” are in Portuguese; check the possibility of translating them into English, as was done for most acronyms in the text.

Line 154: replace “case selection” by “cases selection”.

Line 156 (title of Fig 1): change the acronym “RHC”, which is in Portuguese, to “HCR” (Hospital Cancer Records).

Table 1:

Replace “IRHC” with “IHCR” (Source of the second response variable).

Description of first variable: replace “and standard world population” by “based on standard world population”.

Replace “Explanatory (Contextual)” by “Explanatory Variables (Contextual)”.

In the descriptions of the explanatory variables, for some variables it is described that the population is from the year 2013, and for others it is not. Standardize.

In the description of the variable “Density of family health teams” the term “multiplied by” appears twice.

Consider revising the acronyms “CNES” and “ANS”, if they are changed in the rest of the text.

Line 176: replace “auto-correlation” with “autocorrelation” (no hyphen).

Lines 187-188: rewrite the sentence: “The validation of Moran’s I was carried out by a random permutation test, with 99 permutations” by: “The statistical significance of Moran's I was verified by a random permutation test, with 99 permutations”.

Lines 199-201: replace the phrase: “The multivariate analysis of lung cancer mortality included the explanatory variables that showed a statistically significant correlation with the response variable and non-colinear variables (correlation<0.7).” by: “The multivariate analysis of lung cancer mortality included explanatory variables that showed a statistically significant correlation with the response variable and that were not collinear with each other (correlation < 0.7).”

Line 201: replace “log” with “lag”. The model is called “spatial lag” (not “log”).

Lines 199-205: the authors explain how the multivariate analyzes were performed. In the first review, I had asked “why was multiple analysis done only for mortality, but not for the proportion of lung cancer cases with advanced diagnosis?” - and the authors replied: “The moderate spatial dependence of the proportion of diagnoses in advanced stage of lung cancer (Moran’s I = 0.37) and the weak correlations with the explanatory variables studied (as seen in figures 3 and 4) made it impossible to design a multiple analysis model.” I consider this information relevant to include in the text, perhaps in the limitations item, at the end of the discussion.

5. Results:

Lines 215 and 217: standardize two decimal places in the 95% CI values.

Line 227: replace “19.3%” with “19.36%” (two decimal places).

Lines 230-231: change sentence: “For the combined AMR for men and women, the average was 12.82 (SD 5.12) deaths per 100,000 inhabitants.” by: “The combined AMR for men and women was 12.82 (SD 5.12) deaths per 100,000 population.” (I believe the authors did not average the rates for men and women, which would be wrong, but instead calculated the rate for the entire population of men and women evaluated).

Line 231: where a new sentence starts - “The Itaberaba and Bom Jesus da Lapa ...” – break into another paragraph.

Line 236: “A spatial autocorrelation was observed (Moran’s I 0.50, p=0.01) with the formation ...”– does this autocorrelation refer to AMR? If so, consider rewriting: “AMR presented significative spatial autocorrelation (Moran’s I 0.50, p=0.01); with the formation ...”

Line 255: replace “Figure 4” with “Fig 4”, to keep the same pattern as the titles of the other figures.

Line 259: replace “correlation” with “spatial correlation”.

Lines 260-262: “All correlations are statistically significant, except for correlation with “density of family health teams” (Moran’s I -0.02 p=0.28)” - consider switching to: “All correlations were statistically significant, except for the variable “density of family health teams” (Moran’s I -0.02 p = 0.28)”.

Line 273: change “66%” to “66.3%” and reference Table 2.

Line 275: the variable name – “density of facilities licensed in oncology” is different from the nomenclature of the same variable in Table 2: “Density of Licensed Oncology Health Facilities”. Standardize this nomenclature throughout the text.

Lines 287-289: review the sentence: “The variable Density of Licensed Oncology Health Facilities showed a positive correlation with the studied outcome, although without statistical significance.” to: “The variable Density of Licensed Oncology Health Facilities showed no significant correlation with lung cancer mortality rate”.

6. Discussion:

Line 349: “In Forrest et al. [6]...” – change to: “In Forrest et al. study [6]..."

Line 351: exchange “inequalities social...” – by “social inequalities...”

Line 405: instead of “mainly in territories of Rio Grande do Sul”, change to “mainly in the Rio Grande do Sul state”.

In the first review, I presented some considerations about some divergent results, related to the coverage of health plans and oncology services, comparing the results obtained through the different analysis techniques used. The authors responded brilliantly in the document to the reviewer; therefore, I strongly recommend that such considerations be added, albeit briefly, to the text of the article's discussion. Are they:

Bivariate analysis showed a positive correlation between indicators of health insurance coverage and density of facilities licensed in oncology and the response variable adjusted mortality rates for lung cancer). But it was possible to see, through the maps, that there are clusters of high mortality in regions with low coverage of health plans, especially in IRUAs in the South region.

In the regression model, the density of facilities licensed in oncology lost significance and the coverage of health plans showed an inverse correlation with lung cancer mortality. This discrepancy in results may be related to the levels of socioeconomic inequalities that exist even in the most developed regions, as discussed in the text. This makes access to public or private health services unequal among populations.

Changes in address, in search of treatment, can also move part of the population with higher income to other regions with other health services, including private ones. Likewise, people who need access to health services through the SUS can migrate to places with a greater offer of public services. Both situations can contribute to the reduction of health plan coverage in regions with high mortality rates.

Lines 424-432: regarding this paragraph on the quality of health services, which cites the study by Kaliks et al., I suggest that the authors rewrite the paragraph incorporating their explanation in the document for reviewers, as follows:

The study was carried out with 52 cancer centers linked to the Unified Health System (SUS). The data referring to the treatment were compared to what is recommended by the Ministry of Health, through the Diagnosis and Treatment Guidelines, and to the standard practiced in supplementary health, based on the Procedures List of the National Health Agency (ANS).

Line 431: specify what the abbreviation “SS” means in full.

Line 459: replace the term “can help” by “could help”.

Line 460: replace the term “could have” by “may have”.

7. Conclusions:

Lines 467-469: rewrite the sentence: “The proportion of advanced-stage diagnosis was weakly spatially correlated to socioeconomic and demographics indicators only.” by: “The proportion of advanced-stage diagnoses was weakly spatially correlated only with socioeconomic and demographic indicators.”

Line 471: change the term: “independent of the offer” by “regardless the offer”.

7. PLOS authors have the option to publish the peer review history of their article (what does this mean?). If published, this will include your full peer review and any attached files.

Reviewer #1: **Yes: **Adeylson Guimarães Ribeiro

Reviewer #2: No

Reviewer #3: No

Reviewer #4: **Yes: **Alessandra Cristina Guedes Pellini

---

## [Author Response · Author response to Decision Letter 1]

24 Jan 2022

I commend the authors for the significant improvement related to the new version of their manuscript. However, there are still several issues that must be addressed before it can be reconsidered for publication, as described by the reviewers.

In their first report on this manuscript, reviewer #3 argued that the authors had often misused the term “correlation” in their text. The reviewer claimed that correlation measures do not suggest any direction regarding the relationship between two variables. On the one hand, that explanation is incorrect because correlation coefficients measure the extent to which two variables have a linear relationship and may be positive or negative, thereby indicating the direction of that relationship. On the other hand, the authors responded to the reviewer’s comment by arguing that “The term association is used with qualitative variables. When variables are quantitative, as in the case of this study, the term correlation is the most appropriate. When testing the correlation, the coefficient generated quantifies the strength and direction of the relationship between two numerical variables, with results ranging from −1 to 1, which indicate a positive or negative correlation. Thus, when considering the type of variables included in the study, the term correlation is the most recommended.” Unfortunately, the authors’ argument also has several important flaws. First, it is incorrect to claim that the word “association” should be restricted to the relationship between qualitative variables. Although association and correlation are sometimes loosely used as synonyms, in statistics, correlation is more commonly used to refer to correlation coefficients, whereas association is a more generic term denoting a statistical dependence between two or more variables that may also be positive or negative. Moreover, correlation coefficients reflect the degree of linear association, i.e. the extent to which the points of the scatter plot of two variables are close to a monotonic line but say nothing about the slope of that line, which would be better described the coefficient of a linear regression equation. Hence, for the sake of clarity, I would like to recommend that the authors restrict the use of the term correlation to situations where they refer specifically to correlation coefficients or spatial autocorrelation measures and that they use the term association when referring to the coefficients derived from the equations of bivariate and “multivariate” spatial regression analyses throughout the text. By the way, “multivariable” regression is a more accurate term for the analyses performed in this study than “multivariate” regression. See the following article by Hidalgo and Goodman (https://www.ncbi.nlm.nih.gov/pmc/articles/PMC3518362/)

With regards to the association vs. correlation nomenclature, I recommend checking Miguel Porta’s Dictionary of Epidemiology (https://www.oxfordreference.com/view/10.1093/acref/9780199976720.001.0001/acref-9780199976720). As to the interpretation of correlation coefficients, I would like to suggest the following article by Patrick Schober (https://pubmed.ncbi.nlm.nih.gov/29481436/)

We acknowledge the reviewer’s comments and the terms were corrected.

Reviewer #1: 

All my comments have been addressed and I consider that the manuscript is suitable for publication in this version.

Reviewer #2:

Authors´ affiliation

1. I think there is an error with the affiliation of the first author as in the manuscript appears two different Postgraduate Programs: 'Graduate Program in Collective Health' and 'Graduate Program in Public Health' belonging to the same Department (Public Health or Collective Health?).

The correct affiliation is Department of Collective Health, Postgraduate Programme in Collective Health. It has been corrected.

Abstract

2. I would like to suggest rewriting the sentence 'The adjusted mortality rates for lung cancer presented positive and statistically significant correlation with all the demographic and socioeconomic and health service offer indicators [...].' to 'The adjusted mortality rates for lung cancer presented a positive and statistically significant correlation with all demographic, socioeconomic and healthcare services supply indicators [...].'

The term 'health service offer' appears many times in the manuscript, I would like to suggest considering using 'healthcare services supply' or 'supply of healthcare services'.

We acknowledge the reviewer’s comment and the “health service offer” was replaced by “healthcare services supply” in the manuscript.

3. I would like to suggest to change the term 'Health plan coverage' to 'private health insurance coverage'. The authors sometimes also use the term 'Supplementary Health', I suggest unifying them.

We appreciate the suggestion. The authors chose to unify the term to 'Health plan coverage'.

The description about the origin and calculation of the variable are in the table of variables. 

Introduction

4. I would like to suggest to chance the term 'advanced stage diagnosis' to 'advanced-stage diagnosis' or to 'advanced stage at diagnosis'.

The term has been changed to 'advanced-stage diagnosis'. 

Materials and methods

5. Sometimes, the authors use the term 'Federation Units (FU)' other 'states'. Foreign people do not know if they are the same or not. Please, consider unifying them.

The same problem happens to 'Integrator of Hospital Cancer Records'. Sometimes the authors use 'Integrator' other 'RHC' or 'IRHC' or 'RHC Integrator'. I suggest using the abbreviation 'Integrator-HBCRs' to the term 'Integrator of Hospital-Based Cancer Registries'.

The term been unified to 'Federation Units (FU)' and the abbreviation “ Integrator of Hospital-Based Cancer Registries (Integrator-HBCRs)” was employed.

6. Lung cancer is morphologically classified into different subtypes. Because of this, I suggest changing the term 'carcinoma in situ' to 'in situ disease'.

The text has been changed as required. 

7. Please correct the spelling of 'non-colinear variables' to 'non-collinear variables'.

The text has been changed to:

The multivariable analysis of lung cancer mortality included explanatory variables that showed a statistically significant correlation with the response variable and that were non-collinear with each other (correlation < 0.7).

Results

8. I would like to suggest changing the term 'lung cancer adjusted mortality rates' to 'age-adjusted mortality rates for lung cancer' or to 'age-adjusted lung cancer mortality rates'.

The term has been changed to 'age-adjusted mortality rates for lung cancer’.

Figure 1

9. I would like to suggest changing the term 'without staging' to 'stage not known'.

The term has been modified as requested. 

10. I would like to suggest changing the term 'carcinoma in situ' to 'in situ disease' or to 'not invasive cancer'.

The term has been changed in the text and figure.

11. Please, change the sentence '31,010 (73.2%) cases of lung cancer (age between 18 and 99 years were included)' to '31,010 (73.2%) cases of lung cancer (age between 18 and 99 years) were included'.

The sentence has been modified as required.

S2 Fig

12. Include in the title the term 'Brazil'.

The title has been included. 

S3 Fig

13. Include in the title the term 'Brazil'. Further, include the information about the meaning of

x-axis and y-axis.

Axis names have been changed to full form (Residues for the Spatial Error Model and Lagged Residues for the Spatial Error Model).

Table 1

14. Include in the footnote the meaning of the abbreviations used in the table.

The footnote has been included below the table. 

15. I would like the confirmation of the authors if the description they used for the variable 'adjusted mortality rates for lung cancer' is correct: 'Lung cancer mortality rate, combined for men and women, adjusted for age and standard world population'.

The description has been modified as requested by reviewer 4. 

Reviewer #3: 

In relation to the topic 'correlation' and its use, I was not referring to the measure that varies between -1 and 1 and which is a correlation. The author appears to be using the term correlation when it is a measure of correlation and when it is not. For example, in the title of Figure 2, 'Spatial regression analysis of adjusted lung cancer mortality rates and their correlation with socioeconomic indicators and health service provision, by IRUA, 2011-2015', the author used the term correlation for the coefficients estimated by the regression model, this measure does not vary between -1 and 1, and it is not a correlation. It is more appropriate to use the term association between adjusted mortality and socioeconomic indicator rather than correlation.

Can the author please review?

The text has been reviewed and the terms has been corrected. 

Reviewer #4: 

The manuscript showed substantial improvement after the first review; however, I noticed that some aspects can still be improved. The new suggestions that I present, although they are relevant in number, are easy to resolve.

1. Figures:

Fig. 1: change the acronym “RHC”, which is in Portuguese, for “HCR” (Hospital Cancer Records).

The term has been changed to Integrator of Hospital-Based Cancer Registries (Integrator-HBCRs) as requested.

S1 table: fix the name of the “spatial lag” model (this is described as “spatial log”).

The name has been corrected. 

2.Abstract:

Standardize the number of decimal places. For example: in lines 40 and 41 of the Abstract: “The proportion of advanced-stage diagnosis was 85.28% (95% CI 83.3-87.1).” – The proportion value (85.28%) appears with two places, and the 95% confidence interval with only one (95% CI 83.3-87.1). Standardize (I suggest keeping everything to two decimal places, then getting the CI values right). Also check this in the rest of the text.

The numbers has been standardized as requested.

Line 43: replace “Gini” by “Gini Index”.

The term has been changed. 

Line 45: replace “significant correlation” by “significant spatial correlation”.

The term spatial has been inserted. 

Lines 48 and 52 differ in the denomination of the variable related to Oncology Services – in line 48, it is described as “Density of licensed facilities in oncology”, and in line 52, “Density of Licensed Oncology Health Facilities”. Standardize the term, as well as the rest of the text.

The term has been standardized to density of facilities licensed in oncology.

Lines 51-53: replace the sentence: “The variable Density of Licensed Oncology Health Facilities presented positive correlation with the studied outcome, although without statistical significance.” by: “The variable Density of Licensed Oncology Health Facilities showed no significant correlation with lung cancer mortality rate”.

The sentence has been changed as required.

3. Introduction:

Line 62: the term “neoplasm” still appears in the text. Change the term “neoplasms” to “cancers” (including in reference [1] – GLOBOCAN, it is described as “cancers” and not “neoplasms”).

The term has been changed in the text.

Lines 64-66: “In Brazil, in 2019, there were 16,661 deaths due to lung cancer in men and 12,593 deaths in women, which correspond to an estimated risk of 16.19/100,000 men and 9,84/100,000 women [2]”. First the author presents the numbers of deaths, then he says that this corresponds to estimates of risks (incidences). Review if the data really correspond to the incidence, in this case, do not use the term “which correspond”, but treat the two sentences as independent. Another option would be to present the number of new cancer cases of each sex, and then correlate the respective risks.

The text has been changed to : In Brazil, in 2019, there were 16,661 deaths from lung cancer in men and 12,593 deaths in women, which correspond to an age-adjusted mortality rate of 16.19 per 100,000 men and 9.84 per 100,000 for women [2].

Lines 83-84: replace the phrase: “When considering the socioeconomic and distribution differences of health services in Brazilian regions” by: “When considering socioeconomic differences, as well as the distribution of health services in Brazilian regions”.

The phrase has been replaced as requested.

4. Material and Methods:

I suggest changing the name of the Subsection: “Study design” to “Study design and Spatial analysis units”, since the authors describe, in this item, several aspects related to the analysis units chosen for the study – IRUA.

The name has been changed. 

In the subsection: “Study variables and data sources”, I suggest changing the order of the paragraphs, initially describing everything that refers to the response variables and, later, everything that concerns the exploratory variables. So, the new order would be:

Paragraph #1: “The response variables were the proportion of late-stage diagnosis and adjusted mortality rates ...”

Paragraph #2: “All data included here were collected by municipality and then aggregated to an IRUA territory level by means of averaging ...”

Paragraph #3: “Based on the TNM Classification of malignant tumors, lung cancer cases were classified as advanced stage (TNM III and IV) and early stage (TNM I and II) ...”

Paragraph #4: “The deaths that occurred in Brazil between 2011 and 2015 due to lung câncer (ICD-10, C33-34) [16] were collected from the Mortality Information System (MIS) ...”

Paragraph #5: “For the correction of the number of deaths, the methodology proposed by Santos and Souza [18] was followed ...”

Paragraph #6: “The crude and adjusted mortality rates (AMR) (per 100,000 inhabitants) were calculated for each IRUA ...”

Paragraph #7: “The explanatory variables were classified into demographic and socioeconomic indicators ...”

Paragraph #8: “Demographic and socioeconomic variables for 2010 were obtained from the Brazilian Atlas of Human Development (UNDP) ...” At the end of this last paragraph, keep the sentence already in lines 162-163: “Table 1 presents the study variables and corresponding descriptions ...”

The order of the paragraphs was modified as suggested.

Below the paragraphs of this Subsection, would then come Fig 1 (Flowchart) and Table 1 (Response and exploratory variables).

The figure and table were placed next to the text in which they are mentioned. 

Line 118: describe the term “lung cancer” in italics.

The italic format was used to highlight the expression that was replaced by lung cancer in the text.

Lines 122 and 125: change the acronym “RHC”, which is in Portuguese, to “HCR” (Hospital Cancer Records).

The term has been changed to Integrator of Hospital-Based Cancer Registries (Integrator-HBCRs) as requested.

Line 129: replace the term “for 2011-2015” by “for each year”, as the way it is repeated in relation to line 127, where the years were already mentioned.

The term has been changed.

Line 130: replace the term “residence” by “residence address”.

The term has been changed.

Lines 131 and 133: the excerpt “For the correction of the number of deaths” was repeated. I believe it can be deleted from line 133.

The sentence has been excluded. 

Line 135: does the word “state” mean “Federated Unit”? If so, it would be better to substitute by this last term.

The term has been replaced to Federated Unit (FU).

Lines 135 and 138: replace the acronym “SIM” by “MIS” (as defined this acronym in line 128).

The term has been changed.

Line 146: the acronym UNPD refers to the Organ: “United Nations Development Programme”, which needs to be properly described in full.

The acronym has been corrected.

Lines 147 and 148: the abbreviations “CNES” and “ANS” are in Portuguese; check the possibility of translating them into English, as was done for most acronyms in the text.

The authors chose to keep the abbreviations in Portuguese as they are Brazilian data sources. Access to these sources by readers may be impaired if we translate the abbreviation.

Line 154: replace “case selection” by “cases selection”.

The term has been changed.

Line 156 (title of Fig 1): change the acronym “RHC”, which is in Portuguese, to “HCR” (Hospital Cancer Records).

The acronym has been changed to HBCRs.

Table 1:

Replace “IRHC” with “IHCR” (Source of the second response variable).

The term has been changed to Integrator-HBCRs.

Description of first variable: replace “and standard world population” by “based on standard world population”.

The text has been changed as requested.

Replace “Explanatory (Contextual)” by “Explanatory Variables (Contextual)”.

The term has been changed.

In the descriptions of the explanatory variables, for some variables it is described that the population is from the year 2013, and for others it is not. Standardize.

The table was standardized and the information about the year of the population used in the calculation of all indicators remains in the eighth paragraph of the item "Study variables and data sources".

In the description of the variable “Density of family health teams” the term “multiplied by” appears twice.

The description has been corrected.

Consider revising the acronyms “CNES” and “ANS”, if they are changed in the rest of the text.

The authors chose to keep the abbreviations in Portuguese as they are Brazilian data sources. Access to these sources by readers may be impaired if we translate the abbreviation.

Line 176: replace “auto-correlation” with “autocorrelation” (no hyphen).

The term has been changed.

Lines 187-188: rewrite the sentence: “The validation of Moran’s I was carried out by a random permutation test, with 99 permutations” by: “The statistical significance of Moran's I was verified by a random permutation test, with 99 permutations”.

The sentence has been rewritten.

Lines 199-201: replace the phrase: “The multivariate analysis of lung cancer mortality included the explanatory variables that showed a statistically significant correlation with the response variable and non-colinear variables (correlation<0.7).” by: “The multivariate analysis of lung cancer mortality included explanatory variables that showed a statistically significant correlation with the response variable and that were not collinear with each other (correlation < 0.7).”

The phrase has been changed to: 

The multivariable analysis of lung cancer mortality included explanatory variables that showed a statistically significant association with the response variable and that were not collinear with each other (correlation < 0.7).

Line 201: replace “log” with “lag”. The model is called “spatial lag” (not “log”).

The text has been corrected.

Lines 199-205: the authors explain how the multivariate analyzes were performed. In the first review, I had asked “why was multiple analysis done only for mortality, but not for the proportion of lung cancer cases with advanced diagnosis?” - and the authors replied: “The moderate spatial dependence of the proportion of diagnoses in advanced stage of lung cancer (Moran’s I = 0.37) and the weak correlations with the explanatory variables studied (as seen in figures 3 and 4) made it impossible to design a multiple analysis model.” I consider this information relevant to include in the text, perhaps in the limitations item, at the end of the discussion.

The information has been included in the limitations item.

5. Results:

Lines 215 and 217: standardize two decimal places in the 95% CI values.

The CI has been standardized as requested.

Line 227: replace “19.3%” with “19.36%” (two decimal places).

The information has been replaced to 19.36%”.

Lines 230-231: change sentence: “For the combined AMR for men and women, the average was 12.82 (SD 5.12) deaths per 100,000 inhabitants.” by: “The combined AMR for men and women was 12.82 (SD 5.12) deaths per 100,000 population.” (I believe the authors did not average the rates for men and women, which would be wrong, but instead calculated the rate for the entire population of men and women evaluated).

The sentence has been changed as required.

Line 231: where a new sentence starts - “The Itaberaba and Bom Jesus da Lapa ...” – break into another paragraph.

The text has been modified as suggested.

Line 236: “A spatial autocorrelation was observed (Moran’s I 0.50, p=0.01) with the formation ...”– does this autocorrelation refer to AMR? If so, consider rewriting: “AMR presented significative spatial autocorrelation (Moran’s I 0.50, p=0.01); with the formation ...”

The text has been rewritten as requested.

Line 255: replace “Figure 4” with “Fig 4”, to keep the same pattern as the titles of the other figures.

The term has been changed.

Line 259: replace “correlation” with “spatial correlation”.

The term has been changed.

Lines 260-262: “All correlations are statistically significant, except for correlation with “density of family health teams” (Moran’s I -0.02 p=0.28)” - consider switching to: “All correlations were statistically significant, except for the variable “density of family health teams” (Moran’s I -0.02 p = 0.28)”.

The text has been changed to: “All associations were statistically significant, except for the variable…”

Line 273: change “66%” to “66.3%” and reference Table 2.

The text has been changed.

Line 275: the variable name – “density of facilities licensed in oncology” is different from the nomenclature of the same variable in Table 2: “Density of Licensed Oncology Health Facilities”. Standardize this nomenclature throughout the text.

The term has been standardized to “density of facilities licensed in oncology”.

Lines 287-289: review the sentence: “The variable Density of Licensed Oncology Health Facilities showed a positive correlation with the studied outcome, although without statistical significance.” to: “The variable Density of Licensed Oncology Health Facilities showed no significant correlation with lung cancer mortality rate”.

The sentence has been changed to: 

The variable Density of Facilities Licensed in Oncology showed no significant associated with age-adjusted lung cancer mortality rates.

6. Discussion:

Line 349: “In Forrest et al. [6]...” – change to: “In Forrest et al. study [6]..."

The text has been changed.

Line 351: exchange “inequalities social...” – by “social inequalities...”

The term has been modified as requested.

Line 405: instead of “mainly in territories of Rio Grande do Sul”, change to “mainly in the Rio Grande do Sul state”.

The text has been changed to “mainly in the Rio Grande do Sul FU…”

In the first review, I presented some considerations about some divergent results, related to the coverage of health plans and oncology services, comparing the results obtained through the different analysis techniques used. The authors responded brilliantly in the document to the reviewer; therefore, I strongly recommend that such considerations be added, albeit briefly, to the text of the article's discussion. Are they:

Bivariate analysis showed a positive correlation between indicators of health insurance coverage and density of facilities licensed in oncology and the response variable adjusted mortality rates for lung cancer). But it was possible to see, through the maps, that there are clusters of high mortality in regions with low coverage of health plans, especially in IRUAs in the South region.

In the regression model, the density of facilities licensed in oncology lost significance and the coverage of health plans showed an inverse correlation with lung cancer mortality. This discrepancy in results may be related to the levels of socioeconomic inequalities that exist even in the most developed regions, as discussed in the text. This makes access to public or private health services unequal among populations.

Changes in address, in search of treatment, can also move part of the population with higher income to other regions with other health services, including private ones. Likewise, people who need access to health services through the SUS can migrate to places with a greater offer of public services. Both situations can contribute to the reduction of health plan coverage in regions with high mortality rates.

The considerations were added.

Lines 424-432: regarding this paragraph on the quality of health services, which cites the study by Kaliks et al., I suggest that the authors rewrite the paragraph incorporating their explanation in the document for reviewers, as follows:

The study was carried out with 52 cancer centers linked to the Unified Health System (SUS). The data referring to the treatment were compared to what is recommended by the Ministry of Health, through the Diagnosis and Treatment Guidelines, and to the standard practiced in supplementary health, based on the Procedures List of the National Health Agency (ANS).

The text has been changed to: 

The study by Kaliks et al [51] identified differences in the quality of treatment for the four most common cancers in Brazil, among the Public Health System (SUS), the assistance provided by health plan and the Therapeutic Guidelines established by the Ministry of Health. Of the 52 SUS cancer treatment centers investigated, only 29 followed the directives regarding lung cancer treatment. Eight presented higher standards, five were compatible, and 16 presented lower standards than suggested by the Therapeutic Directives. Furthermore, 19 SUS cancer centers presented standards below the recommended and practiced by health plans.

Line 431: specify what the abbreviation “SS” means in full.

The correct term is health plan.

Line 459: replace the term “can help” by “could help”.

The term has been changed.

Line 460: replace the term “could have” by “may have”.

The term has been changed.

7.Conclusions:

Lines 467-469: rewrite the sentence: “The proportion of advanced-stage diagnosis was weakly spatially correlated to socioeconomic and demographics indicators only.” by: “The proportion of advanced-stage diagnoses was weakly spatially correlated only with socioeconomic and demographic indicators.”

The text has been changed to: 

A high proportion of advanced-stage lung cancer diagnoses was verified for almost the entire Brazilian territory for the study period. The proportion of advanced-stage diagnoses was weakly spatially associated only with socioeconomic and demographic indicators. However, age-adjusted mortality rates for lung cancer presented irregular distribution in the Brazilian IRUA, associated with territories that present higher incomes and lower coverage of health plans, regardless the supply of health facilities licensed in oncology.

Line 471: change the term: “independent of the offer” by “regardless the offer”.

The term has been modified as sugge

---

## [Editor Report · Decision Letter 2]

1 Mar 2022

Spatial assessment of advanced-stage diagnosis and lung cancer mortality in Brazil

PONE-D-21-20112R2

Dear Dr. de Souza,

We’re pleased to inform you that your manuscript has been judged scientifically suitable for publication and will be formally accepted for publication once it meets all outstanding technical requirements.

Kind regards,

Edison I.O. Vidal, MD, MPH, PhD

Section Editor

PLOS ONE
---

## [Editor Report · Acceptance letter]

10 Mar 2022

PONE-D-21-20112R2 

Spatial assessment of advanced-stage diagnosis and lung cancer mortality in Brazil 

Dear Dr. de Souza:

I'm pleased to inform you that your manuscript has been deemed suitable for publication in PLOS ONE. Congratulations! Your manuscript is now with our production department. 

Kind regards, 

on behalf of

Professor Edison I.O. Vidal 

Section Editor

PLOS ONE